# Delineating the Technosphere: Definition, categorization and characteristics

Eric Galbraith[1,2], Abdullah Al Faisal[1], Tanya Matitia[1], William Fajzel[1], Ian Hatton[1], Helmut Haberl[3], Fridolin Krausmann[3], and Dominik Wiedenhofer[3]

[1]Earth and Planetary Sciences, McGill University, Montreal, Canada
[1]Institut de Ciencia i Tecnologia Ambientals (ICTA-UAB), Universitat Autonoma de Barcelona, Cerdanyola del Valles, Spain
[3]Institute of Social Ecology, BOKU University, Vienna, Austria

**Correspondence:** Eric Galbraith (eric.galbraith@mcgill.ca)

**Abstract.** The global assemblage of human-created buildings, infrastructure, machinery and other artifacts has been called the 'technosphere', and plays a major role in the present-day dynamics of the Earth system. The technosphere enables the rapid extraction of natural resources and the combustion of fossil fuels, impacting biodiversity and causing climate change, while generating copious amounts of waste materials. At the same time, the technosphere supports humans in many ways, including the provision of food, shelter, transportation and long-distance communication, and it is the main component of material wealth. Despite its importance, Earth system science has been slow to explicitly incorporate the technosphere as an integrated part of its conceptual and quantitative frameworks. Here we propose a refined definition of the technosphere, intended to assist in developing functional integration with other Earth system spheres as well as social sciences. We also suggest a categorization system for the things that make up the technosphere, based on how their end-uses support human motivations. Given the formal definition and resolved categorization, we delineate basic attributes of the technosphere, including its mass distribution among categories and across the Earth surface, and discuss its first-order temporal dynamics. In particular, of the 1 trillion tonne technosphere mass, we estimate that roughly one-half is buildings and one-third transportation infrastructure, both of which we map globally at one-degree resolution. Movable entities, mostly composed of vehicles, vessels, and machinery, account for less than 2% of the total technosphere mass, yet are comparable to the biomass of all animals on Earth. We show that reconstructions of the technosphere since 1900 are consistent with an autocatalytic process, resulting in exponential growth with long-run increase of $>3\%$ y$^{-1}$, equivalent to a 20-year doubling time. Building a stronger quantitative understanding of the technosphere can help to better integrate it within Earth system science, while bridging natural and social sciences to support physically-plausible pathways towards sustainability and human wellbeing.

## 1 Introduction

More than 8 billion humans live on Earth, embedded within a massive network of roads, buildings, vehicles, machinery, computers and other artifacts. The entirety of these human creations, arrayed across the surface of our planet as well as orbiting above it, has been referred to as the 'technosphere' (Haff, 2023). Humans create the technosphere to provide diverse end uses such as transport, comfortable living spaces and communication, as well as for social reasons (Pauliuk and Müller,

2014; Schaffartzik et al., 2021). We are also entirely dependent on it to keep us alive - no more than a few million humans
could be provided with food, drinking water, or shelter without it (Daily and Ehrlich, 1992; Fischer-Kowalski and Weisz, 1999; Smil, 2004).

The use of the term technosphere has been justified by the fact that the assemblage of human creations can be considered a large and complex functional aggregate, like other spheres of the Earth system (Table 1). Like the atmosphere, in which convection at one location can be strongly linked to precipitation at a distant location a week later, components of the technosphere
are involved in many globally-interconnected internal processes. For example, the functions of mining machinery are strongly linked to transportation networks, metal processing facilities and manufacturing plants. Like the biosphere, the technosphere supports its own growth over time by increasing the rate at which materials can be extracted, processed, transported, and transformed to final products (Herrmann-Pillath, 2018). In addition, it represents a significant and rapidly changing component of the Earth surface. Depending on how it is defined, the technosphere has been estimated to be of comparable scale to the
biosphere, both in terms of its mass and the fluxes it enables. For example, the mass of human creations likely exceeds the dry mass of all living organisms (roughly 1100 Gt dry mass) (Elhacham et al., 2020), the rate of primary energy conversion within the technosphere ($\approx$ 20 TW) is roughly half that of terrestrial above-ground net primary production (Haberl et al., 2007) and continues to increase (Smil, 1991; Lenton, 2016), and the rate of mass dislocation at the Earth surface by machines (roughly 320 Gt y$^{-1}$) appears to exceed all natural geomorphological processes by an order of magnitude (Cooper et al., 2018). There
seems little doubt that, at this point, the technosphere can be considered a major component of the Earth system.

The technosphere is also central to two prominent themes of discussion in the realm of Earth system science and sustainability: planetary boundaries (Steffen et al., 2015) and the wellbeing of humans (Stiglitz et al., 2009; Raworth, 2018). The vast scale and complexity of the technosphere make its nature difficult to grasp, and like the proverbial fish who is unaware of the water she swims in, we can be remarkably oblivious to the role of technosphere dynamics in both of these themes. By trying to
45 make sense of the whole technosphere, at the planetary scale and in connection with human lives, we can better comprehend how and why it comes into existence, as well as its functional role in driving global change.

Yet despite its importance, the technosphere remains absent from most conceptions of Earth system science (Herrmann-Pillath, 2018). The term is inconsistently defined, lacks a system of categorization, and its basic attributes have not been holistically assessed, including its mass distribution and temporal dynamics. Industrial ecology and ecological economics have
50 made great strides in estimating the fluxes of material and energy through the global technosphere, under the names industrial metabolism or socioeconomic metabolism research (Graedel et al., 2015; Weisz et al., 2015; Pauliuk and Hertwich, 2015; Haberl et al., 2019; Lanau et al., 2019; Fu et al., 2022). However, so far there have been few efforts to integrate this work with the Earth system science perspectives. In short, the understanding of what the technosphere is, as an Earth system component, remains remarkably poorly resolved.
This paper aims to improve the resolution of the technosphere by providing an interdisciplinary foundation for linking its material basis with its functionality, and by presenting a compilation of data that gives some insights on its geographical and dynamical characteristics. The paper is structured as follows. Section 2 reviews existing definitions of the technosphere and proposes a refinement. Section 3 presents a descriptive categorization scheme for technosphere entities, aligned with the human

motivations that underlie their creation. Section 4 assesses the composition of the technosphere in terms of the categorization,
and provides maps of its first-order distribution across the Earth surface. Section 5 discusses basic empirical features of the
temporal dynamics of the technosphere, focusing on its catalytic properties, and Section 6 concludes the paper.

## 2   Defining the boundaries of the technosphere

The term 'technosphere' has been attributed to science writer Wil Lepkowski, who was apparently the first to use it in a 1960
article (Otter, 2022). It was subsequently used by systems engineer John Milsum (Milsum, 1968) and, the following year, by
65 biologist Julian Huxley in a reflection on the first moon landing (Huxley and Nicholson, 1969). More recently, geologist Peter
Haff effectively promoted the term to encapsulate the global proliferation of human technology at the heart of the concept of the
Anthropocene (Haff, 2014, 2023). Haff described the technosphere as including "everything that enables rapid extraction from
the Earth of large quantities of free energy, long-distance, nearly instantaneous communication, rapid long-distance energy
and mass transport, high-intensity industrial and manufacturing operations, and a myriad additional 'artificial' or 'non-natural'
processes without which modern civilization could not exist". His use of 'technosphere' was chosen rather than 'anthropo-
sphere' to suggest a detached view of an emerging geological process that has partly entrained humans, rather than one that has
humans exclusively at the centre. Zalasiewicz (2017) used a slightly modified version of this definition, specifying the part of
the technosphere which is currently in use. The in-use portion of the technosphere is also distinguished by Johansson (2013),
and is generally equivalent to commonly used terms in industrial ecology, including 'in-use stocks' (Pauliuk and Hertwich,
2015), 'material stocks' (Haberl et al., 2019), 'manufactured capital' (Weisz et al., 2015) and 'technomass' (Inostroza, 2014),
as well as 'artefacts' in the early socio-ecological literature drawing on ecological anthropology (Fischer-Kowalski and Weisz,
1999). Throughout the years, the technosphere has been defined in many different ways, sometimes including human-disturbed
soils, social processes, or even ideas.

Although there can be no 'correct' definition of the technosphere, we propose here a refinement which is compatible with
80 the standard Earth system spheres conceptualization (Table 1). We follow the biogeochemist's convention of defining a sphere
in terms of the properties of the matter of which it is comprised (i.e. stocks), rather than in terms of processes (i.e. fluxes). In
other words, to use the terminology common among industrial ecologists and economists, a sphere is defined as an assemblage
of stocks, not according to flows. For example, the atmosphere is defined as the envelope of gas that surrounds our planet,
including tiny particles suspended within it. There are many processes that occur within the atmosphere, such as convection,
precipitation and cloud formation, but these are not what define the atmosphere. Similarly, the biosphere is defined as the stock
of living organic matter, rather than the processes and flows that carry on the business of life, although we recognize that many
authors do take a more expansive definition (e.g. Folke et al., 2011).

Even with a deliberate focus on stocks, the boundaries of the technosphere are inherently blurry. To varying extents, this
bluriness applies to all spheres of the Earth system. For example, one could ask whether bubbles of air mixed into the surface
ocean by waves belong to the atmosphere or the hydrosphere, or if they oscillate back and forth as they are injected and
subsequently outgas. As such, the sphere framework is not obviously suited to precise categorizations, at least not without a

| Sphere | Description |
| --- | --- |
| **Lithosphere** | The rigid planetary crust, composed of a continuous spheroid of rock. |
| **Regosphere** | The mixture of non-living debris that lies on top of the lithosphere, including the inorganic and organic components of soils and marine sediments. |
| **Atmosphere** | The layer of gas that envelopes the Earth, including suspended solutes and particles. |
| **Hydrosphere** | Liquid and solid phases of water comprising the ocean, lakes and rivers, groundwater, permafrost, snowpack and glaciers, including solutes and entrained particles. |
| **Biosphere** | All living organisms, from unicellular bacteria to whales, including humans and domesticated animals. |
| **Technosphere** | All non-food matter extracted from other spheres of the Earth system and transformed to novel states that can provide end-uses intended by humans. |

**Table 1.** Conceptualizing the spheres of the Earth system, including the technosphere, defined according to the constituent matter. See Huggett, (2024) for a discussion of the spheres.

long list of instructions on how to treat edge cases. Nonetheless, the spheres have proven helpful for sketching, in broad strokes, the components of our planetary system in an intuitive way, helping to think more clearly about processes that extend up to the global scale.

To these ends, the technosphere is here defined as: *all non-food matter extracted from other spheres of the Earth system and transformed to novel states that can provide end-uses to humans*. We highlight four important distinctions inherent in this definition.

 First, we limit the components of the technosphere to nonliving creations, i.e. not including organisms composed of cells with active ribosomes. As such we do not include living humans or any other life form within the technosphere, but instead

consider all living organisms as components of the biosphere (where the biosphere is the sum of all organisms). This provides long-term continuity with Earth history, since humans were clearly a part of the biosphere in the ancient past, as were the ancestors of our domestic animals, and there was no point at which we 'left' the biosphere. Our food continues to be derived almost entirely from the living organisms that comprise the biosphere, with whom we also share viruses and bacteria. Thus, the definition here considers broiler chickens, grapefruit, oil palms, corn and genetically-modified sheep as part of the biosphere,

while pacemakers and prosthetic limbs are considered part of the technosphere. Also, we do not include modifications of the regosphere or lithosphere as part of the technosphere, thus excluding the soils of croplands and rubble of mines, all of which would remain classified in the regosphere. This definition avoids conflating the technosphere with the meaning of 'artificial', a conflation which easily occurs with the term 'anthroposphere' (Pauliuk and Hertwich, 2015) and which promulgates a false dichotomy between natural and artificial worlds. Because of these exclusions, the technosphere definition proposed here implies

a much smaller mass than that estimated using the definition proposed by Zalasiewicz (2017), which was dominated by human-disrupted soils and sediments.

Second, because this definition refers exclusively to non-living physical matter, it does not include human activities or immaterial social constructs like institutions, corporations, or social norms. In physical terms, social processes are couched in the neural structures of humans, coordinated by symbolic information exchange, and neurons are located within the biosphere (Galbraith, 2021). This is not to say that the technosphere is independent from social dynamics. Rather, social processes and the technosphere are strongly coupled, in the same way that the growth of plankton in a culture depends on the nutrient content of the water in which they grow, or the dynamics of the ocean and atmosphere are tied through the exchange of energy. In addition to providing a more unambiguous definition, this separation of human society from the technosphere preserves the independence of human agency, which was identified by Donges (2017) as a conceptual problem with the technosphere of Haff (2014).

Third, we draw a boundary where the state of an entity deteriorates so as to be unfit to serve an intended end-use. An object that undergoes repair remains in the technosphere, while an object that is discarded or irreparably damaged does not. As such, this boundary can be subject to social characteristics that determine the willingness or ability to maintain or repurpose items. The out-of-use boundary can be seen as analogous to the definition of the biosphere as comprised exclusively of living organisms – when an organism dies, it ceases to be part of the biosphere. And, just as heterotrophic consumption can recycle organic matter within the biosphere, material recycling can transform defunct technosphere components into new in-use entities (see recycling flux in Figure 1). Similarly, a medieval fortress that was abandoned after losing its original functionality can be repaired and returned to the in-use technosphere as a museum.

Fourth, this definition of the technosphere does not include mass that would be called 'waste'. Although dealing with waste is clearly important for humans, sustainable development, and the Earth system, a precise definition that works on long timescales is difficult to construct. Most of what would be considered technosphere waste is gradually transformed or mixed across the Earth system on timescales from hours to millennia, making it difficult to define a boundary at which it would ever stop being waste. This differentiates the technosphere definition proposed herein from the anthroposphere concept, which would include all waste ever produced by humanity (Pauliuk and Hertwich, 2015). Furthermore, biological analogs of technosphere waste are not typically considered part of the biosphere in Earth system science. For example, exhaled $CO_2$ is part of the atmosphere, while dissolved organic molecules in seawater are part of the hydrosphere, and the carbonate shells of foraminifera accumulated in sediments are part of the regosphere. Placing the boundary in this way captures the fact that abandoned components of the technosphere are inexorably re-incorporated into the other spheres, rather than remaining apart. A discarded soda can interacts with its surroundings as part of the regosphere, microplastics suspended in the ocean are part of the hydrosphere, and polychlorinated biphenyls incorporated into living organisms are part of the biosphere. Although external to the technosphere, persistently identifiable wastes can be termed 'technofossils', as suggested by Zalasiewicz (2014). Thus, the flux of matter through the technosphere has introduced many novel chemicals and structures that are now distributed throughout the other spheres of the Earth system, and have fundamentally changed our planet, just as the flux of oxygen from the biosphere changed the redox state of the atmosphere billions of years ago (Lenton, 2016).

As illustrated in Figure 1, we distinguish the in-use portion of the technosphere from substances that have been extracted or produced from the other Earth system spheres, but remain in an intermediate state as materials or components. We also

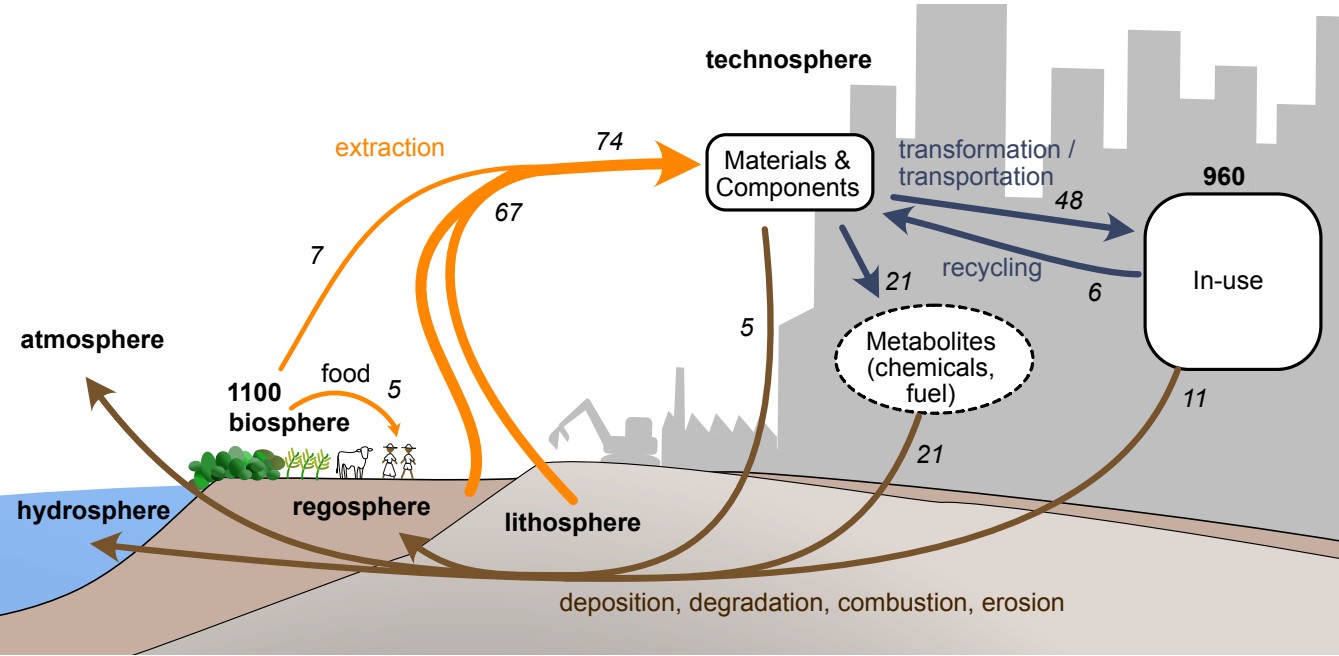

**Figure 1.** Schematic of the technosphere and its exchanges within the Earth system. Italicized numbers show fluxes in $Gt \ y^{-1}$ and the in-use mass is shown in bold in $Gt$, as estimated by Krausmann et al. (2018) for the year 2015. For comparison, the dry mass of the biosphere as estimated by Elhacham et al. (2020) is shown in bold, and the dry mass of food flux estimated by Alexander et al. (2017) is shown in italicst.

separate substances that are chemically transformed, in a single use, to an unusable state, and refer to these as technosphere metabolites, by analogy with the molecules that are transformed within organisms to provide metabolic energy and fuel growth. The technosphere metabolites include fuels (fossil fuels, firewood, bio-ethanol, uranium ore), industrial chemicals (reagents,

fertilizers and pesticides) and pharmeceuticals. Food – the organic matter produced by organismal growth and consumed by living humans – is not considered part of the technosphere.

We also note that this definition of the technosphere is equivalent to the concept of in-use material stocks of socio-ecological metabolism research (Fischer-Kowalski and Weisz, 1999; Haberl et al., 2019) while excluding human bodies and domesticated animals (since the Earth system framework places all of these within the biosphere). This provides a consistency with economy-

wide Material Flow Accounting (ew-MFA, see below) which is used in Figure 1 to estimate fluxes within the technosphere and with other spheres of the Earth system (Krausmann et al., 2017a; UNEP, 2023).

## 3   Categorizing the technosphere

It is clear that the technosphere is huge, important, and dynamic. But what is this global construct that surrounds us? A systematic and holistic categorization of the technosphere can provide a global scale perspective, making it easier to comprehend what

the whole technosphere is composed of, as well as contextualizing the parts within it. It can also help to build an understanding of the functional behaviour of the technosphere, and how it relates to outcomes for human wellbeing.

However, developing such a categorization is not an easy task, nor can it be seen to have a unique solution. The technosphere is extremely diverse in composition, function, and the interactive roles it plays within human societies. It also evolves over time, often in coordination with social changes, so that designing a categorization that is robust on long timescales is challenging.

Nonetheless, even imperfect classifications can provide useful frameworks within which to better understand a system, just as the Linnean classification of species (Linnaeus, 1758) continues to prove a useful framework for the biosphere, despite the fact that it cannot ever capture the underlying reality of phylogeny (Benton, 2000). The remainder of this section develops one such categorization for describing the components that make up the technosphere, aligned with the outcomes that motivate their creation.

## 170  3.1  Existing categorizations

Extensive classification systems of human creations do already exist, largely developed by economists and business entities to organize commerce and develop trade statistics. For example, the central product classification (CPC), developed by the United Nations (UNSD, 2015) provides an exhaustive and exclusive categorization of goods and services – the outputs of economic activity. The CPC consists of more than 4000 classes, nested within a 5-level hierarchical classification. However, 175 these categories are designed to apply to publicly-available economic data, rather than a physical systems understanding of technosphere function within the human-Earth system. As a result, many classes are specific to materials or manufacturing sectors, or include intermediate components, and these are frequently mixed together. For example, one category includes "Medical appliances, precision and optical instruments, watches and clocks", which aggregate a wide range of end uses based on the technical nature of manufacturing.

A more Earth-system relevant set of categorizations has been used within the framework of economy-wide Material Flow Accounting (ew-MFA), which has been developed for the purpose of monitoring the biophysical basis of society and informing sustainable resource use policies (Krausmann et al., 2017a; UNEP, 2023). The ew-MFA framework is widely used to provide policy-relevant indicators in the Sustainable Development Goals and for national resource policies, consistently accounting for all fluxes and in-use stocks in physical units. This accounting draws on data using the CPC classifications and other socio- 185 economic statistical data sources such as the Food and Agriculture Organization, International Energy Agency and United States Geological Survey. Annual raw material extraction and physical trade between economies are reported, as well as the material footprint, i.e. all upstream material use along supply chains whose products are ultimately destined for consumption elsewhere (Wiedmann et al., 2015; Lenzen et al., 2022; UNEP). Work within this framework has primarily categorized fluxes and in-use stocks by their material properties, or focused on specific end uses within a subset of the technosphere (Chen and 190 Graedel, 2015; Lanau et al., 2019; Streeck et al., 2023). Recent developments in ew-MFA have provided the first fully mass-balanced accounts of the global socio-metabolic system including raw material extraction and in-use material stocks, as well as all waste and emissions (Krausmann et al., 2017b, 2018; Wiedenhofer et al., 2019). Economy-wide, fully mass-balanced

accounts across all bulk materials, which can be linked to their main end-uses, have become available even more recently (Wiedenhofer et al., 2024b) which we will return to below as a key component of our data compilation.

## 3.2 Alignment with intended purpose

We seek an end-use categorization that can help to reveal functional aspects of the technosphere, both in its relations with human societies and the rest of the Earth system. In the interest of avoiding ambiguity, one might imagine starting directly from the measurable physical changes in the world caused by the use or operation of a technosphere entity. For example, some entities can be used to generate heat, or transfer heat from one place to another. However, if we want to link to the reasons that an entity is produced, or to the benefits it provides, this approach would be very indirect. For example, heat might be generated in a smelter to produce nickel, in a kettle to make tea, or in the boiler of a steamship to power transportation: the immediate physical change is similar, but the desired outcomes that motivate the heat production are very different. A similar problem arises if we consider the socially-mediated reason to produce something, such as the desire for an expensive car as a status symbol, rather than as a means of transportation. Although this may be very important in terms of societal dynamics, the same object can have different social significance at different times and in different cultures, making it hard to use as a basis for categorization.

We therefore choose to focus on physically-oriented end-use outcomes that are aligned with human motivations. End-use outcomes are essential aspects of the technosphere, since they are are what cause humans to create it - every component of the technosphere can be associated with at least one type of intended practical end use. Note that the word 'motivation' is used here in a very general sense, rather than in the sense of psychological theories of motivation (e.g. Maslow (1943)), and could substitute it with the word 'purpose'. Many end-uses can be mechanistically linked to physical, material impacts on the Earth system, including energy transformations, mass transport, biosphere modification, and chemical processes. Other end uses change our surroundings, altering the context in which we spend our time. Thus, a motivation-aligned end-use perspective connects naturally to both human purpose and Earth system outcomes.

We also use the fact that many technosphere entities operate in direct support of specific human activities in a way that materially alters their outcomes. This key aspect of the technosphere can amplify human impacts on other parts of the Earth system in a way that is tied to human time allocation. For example, the ability to catch fish is enhanced by the use of a fishing boat and fishing gear, while a highway increases the rate at which people and goods can be transported. Activities can be defined in many ways, but to facilitate quantitative connections with existing data, we align the end uses with activities from the Motivating Outcome-Oriented General Activity Lexicon (Galbraith et al., 2022; Fajzel et al., 2023) wherever feasible. This categorization was created to provide an exhaustive and exclusive categorization of human activities based on the outcomes that motivate the undertaking of the activities. Outcome orientations were identified in published economic, anthropological and sociological categorizations, and harmonized using physically-based descriptions to clearly identify processes that alter the Earth system in material terms. The alignment with activities provides the benefit of clearly associating the technosphere with the human activities with which they are engaged, at least for the components for which this relationship is clear.

## 3.3 MEUTEC end-use categories

Table S1 provides an overview of the Motivating End-Use Technosphere Entity Categorization, or MEUTEC for short. The MEUTEC is intended to provide an exhaustive and exclusive set of categories for the in-use technosphere, i.e. extracted materials that have been transformed into the state intended for use, and persist in a usable state. The MEUTEC is shown alongside the activity classification in Figure 2. Shaded rectangles indicate connections between groups of activities and technosphere categories, reflecting how the purposes of technosphere entities can be connected to particular activities, as well as altering the outcomes of activities. For example, the MEUTEC category 'Energy extraction & conversion' includes entities that exist to support the 'Energy' activity, and whose use enable the provision of energy at greater rates, and in different forms, for a given investment of activity.

In contrast, many parts of the technosphere, shown near the bottom of Figure 2, are not associated with one of the specific activities. These entities are created to change the physical context in which human bodies exist, such as the buildings that serve as our ambient environments, our clothing, and furniture. We also include here the infrastructures that control the flow of water in our surroundings. Together, we refer to these as the Ambient context category.

We highlight a broad array of activity-associated entities that are intensely relevant to the functioning of the global human system, but are particularly difficult to categorize. We refer to this overall group as *Information, organization and neural interaction*. Many of these entities include devices, artifacts and infrastructure to provide neural stimulus, rather than making physical changes in the external world. We also include here diverse entities that help organize social behaviour, including entities that help organize trade, and military equipment. These diverse entities are hugely important for society and collective cognitive processes, acting in complex and interconnected ways.

A particularly important subset of entities store, process, generate or display symbolic information – physical patterns that may be written words and numbers, or digital (electrical impulses representing strings of zeros and ones) and referred to as Information and Communication Technologies by Creutzig et al. (2022). These entities can provide high quality communication (Boyd, 2018), preserve information over time, externalize thought processes, and enable complex logical operations and computation. Information can be stored in many forms, including as engravings on stelae, words on paper, or magnetic fields in solid-state electronics. Long-distance transmission is enabled by radio antennae, fiber optic cables, and communications satellites while computers are particularly important for processing of information in the modern world.

Also among the difficult-to-categorize entities are many that do not utilize symbolic information, such as dice games, musical instruments, and photographs. These were all invented before the digital age, but all can now be replicated or simulated with computers, and transmitted interactively using the internet. This convergence of purpose across material technologies may reflect the underlying importance of information-rich patterns to the forms of neural stimulus that we find both useful and pleasurable.

The connections indicated within the Information, organization and neural interaction categories should be seen as provisional, and we hope that they can be improved in future work.

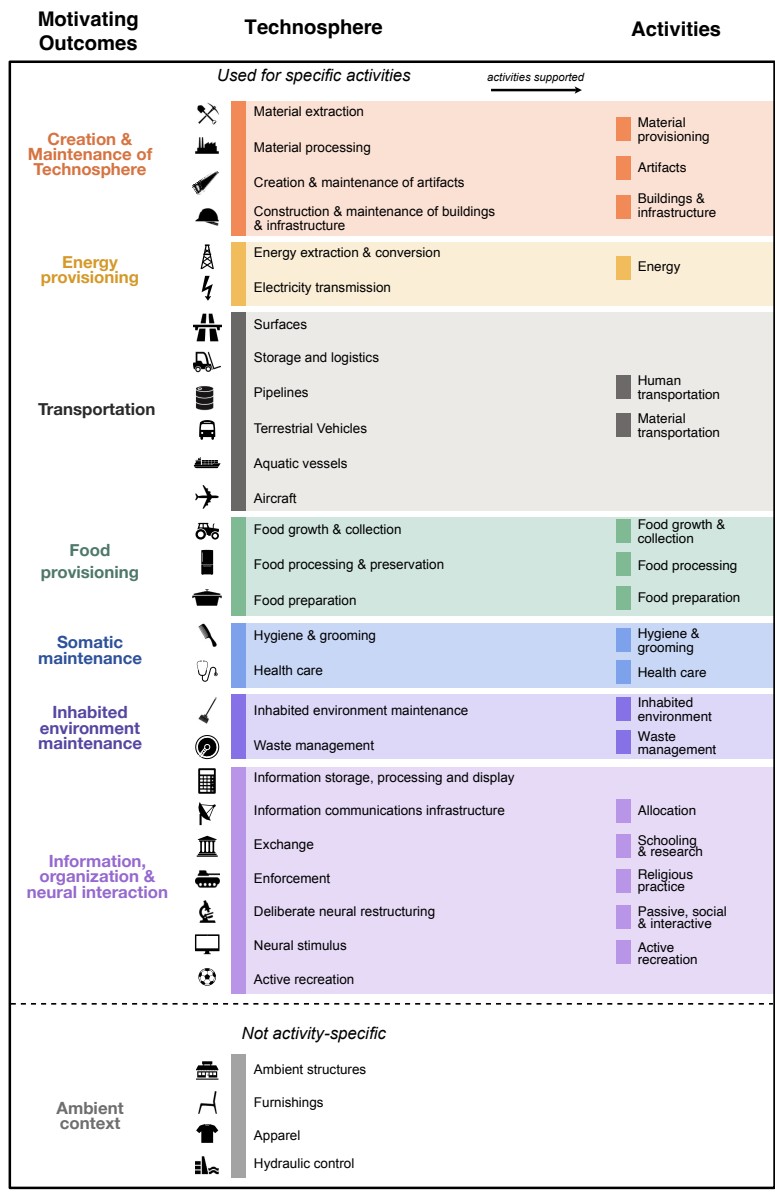

**Figure 2.** Motivating end-use categorization of the technosphere. The upper portion shows categories of entities that support specific activities. Technosphere categories that do not have clear associations to specific activities, but instead modify the ambient context in which humans exist, are shown below.

## 3.4 Assigning entities to categories

Most of the identifiable objects/entities of the technosphere are themselves assembled from multiple components, what Oswalt (1987) calls technounits. For the purpose of categorization, we generally consider an entity at the scale at which the motivating end use is fulfilled, which one could regard philosophically as a type of holon. A holon, *sensu* Koestler (1970), is recognizable as a distinct and functional unit, yet it is composed of multiple parts, while itself being part of a larger assemblage. Thus, an *entity* is here considered as a coherent spatially-organized and persistent object that provides an end use (i.e. a service) without

requiring additional components to do so, although it may require an input of metabolites and/or energy. Thus, a building is an entity, including all integral plumbing, wiring, paint, and exterior cladding, while a chair inside the building is a separate entity (since it would provide the same end use if placed outdoors, though its occupant might get rained on). In addition, the assignment of entities to categories applies the following principles:

– **Outcome-oriented.** The relevant end-use is the physical outcome which motivated the production of the entity or its

ongoing maintenance. For example, the creation of a television is motivated by the desire to provide experiences to viewers, rather than a desire to convert electrical energy to electromagnetic radiation for its own sake. Superficially similar entities may sometimes fall into different categories. For example, a hunting knife would be categorized in food growth and collection, while a kitchen knife would fall into food preparation. The motivating outcome is determined by the apparent reason for investment of human activity and/or energy in its creation/maintenance.

– **Avoid non-experiential outcome.** Because any interaction with an entity will change the context of human experience, this is not used as the basis of categorization unless there is no other clear motivating outcome. For example, a car can provide a pleasant context for sitting and listening to music at a comfortable temperature, but it is the primary physical purpose of transporting humans and goods that is used for categorizing the car.

– **Avoid social significance.** The social, cultural, or economic significance of an entity is not considered, unless there is no

other end-use. Instead the classification is according to the ostensible, physically-grounded end use. For example, a block of apartments would be categorized as residential buildings, regardless of whether their construction was motivated by an actual need for housing or by capital investment strategies.

– **Material-agnostic.** The material of which an entity is comprised does not influence its classification. A jacket is classified in the same way, whether the material is derived of petroleum (nylon), plants (cotton), or animals (leather). Electron-

ics are considered a particular collection of materials, rather than an end use. A single category can include both fixed, immovable creations (i.e. elements of the built environment) as well as movable entities, where they both contribute to the same type of outcome. For example, the end use 'material processing' can include machinery as well as constructed refining facilities.

– **No components.** Components are not considered as entities. For example, a screw is a component, which could become

part of an end-use entity by being incorporated in a residential building, or in furniture. Similarly, engines are not classified as entities, but included within the vehicle they power.

- **Priority scheme.** To reduce ambiguity among entities that could equally fall in more than one category, we define priority rules (Supplementary Table S1).

## 3.5 Application to existing lists of entities

Over long timescales, past and future, significant cultural changes and innovations are likely. Although we cannot know what the future will bring, we can demonstrate the applicability of MEUTEC categories to available lists of human creations from different cultures.

First we apply the categorization to lists compiled by ethnographers for two hunter-gatherer societies, as reported by Kelly (2013). The Ju/'huan live in the sub-tropical Kalahari Desert, while the Nuvugmiut live at Point Barrow on Alaska's northern coast. As shown in Table 2, the 20 entities of the Ju/'hoansi technology and 36 entities of the Nuvugmiut technology are all readily associated with one of the level 2 MEUTEC categories. Food provisioning is the most common level 1 category for both societies, according to the way the ethnographers described item types. Within the Nuvugmiut technology, both the Creation & Maintenance of Technosphere and the Transportation entities are more common than they are in the Ju/'hoansi technology, consistent with a greater reliance on the technosphere to persist in the cold environment and to undertake long-distance travel over snow, ice and water. Notably, four level 1 categories are unrepresented in both societies. It is possible that some artifacts that would have fallen in these categories (such as combs and dice games) were not recorded by the ethnographers. However, the absence of these categories is also consistent with their development as a hallmark of larger societies in which labour specialization is more prominent (Graeber and Wengrow, 2021).

We also applied the MEUTEC categorization to the Central Product Classification (UNSD, 2015), mentioned above, after removing services, raw materials, components and metabolites. This resulted in 389 classes of finished, traded goods, which were associated with MEUTEC categories as listed in Supplementary Table S2. Unlike with the hunter-gatherers, every MEUTEC category had representative goods in the CPC, from a minimum of 2 (Electricity transmission) to a maximum of 45 (Apparel). Some classes were difficult to uniquely categorize, particularly those related to information and neural stimulus, but these represented a minority.

The fact that the MEUTEC can be used with these two very different types of technosphere entities is encouraging for its potential application to long timescales, both for historical changes and future projections. Nonetheless, the fact that prominent categories in modern industrialized society were apparently minor or absent among hunter gatherers is a good reminder that new categories may be required in the future. For example, general-purpose robots capable of doing most human tasks would not easily fit into any existing categories.

## 4 Basic attributes of the technosphere

Given the formal definition and categorization adopted above, we now characterize basic features of the technosphere. We first provide an overview of how the mass of the technosphere is partitioned among the MEUTEC categories and across the planet surface, circa year 2019. The overall mass of technosphere components is not necessarily the most relevant variable for

| Level 1 | Level 2 | Ju/'huan | Nuvugmiut |
|---|---|---|---|
| Creation & maintenance of technosphere | Creation of artifacts | adze | mauls, adzes, chisels, saws, awls, whetstones, scrapers |
| Energy provisioning | Energy extraction and converters | fire-making equipment | bow drills |
| Transportation | Storage and logistics | carrying bags, ostrich egg canteens | wooden pails, storage boxes |
| | Terrestrial vehicles | - | sledges |
| | Aquatic vessels | - | kayaks, umiaks |
| Food provisioning | Food growth & collection | bow, arrows, quiver, spear, throwing stick, springhare pole, carrying net | fishhooks, sinkers, fishing line, leisters, fishing nets, bows, arrows, quiver, atlatl, bola, snares, harpoons |
| | Food processing & preservation | nut-cracking stones | - |
| | Food preparation | knife, bowls, spoons, mortar and pestle | wooden bowls, knives, dippers, spoons, ladles |
| Inhabited environment maintenance | - | - | - |
| Somatic maintenance | - | - | - |
| Information, organization & neural interaction | - | - | - |
| Ambient context | Ambient structures | hut | house |
| | Furnishings | - | soapstone lamps |
| | Apparel | clothing, bead ornaments | clothing, goggles, mittens |

**Table 2.** Categorization of hunter-gatherer material cultures. Ethnographic lists of entities were taken from Kelly (2013). Each entity was associated with a MEUTEC Level 2 category (unused Level 2 categories are not shown).

Earth system interactions – for example, the extraction and processing of a kilogram of gold can have far greater environmental impacts than the extraction of a kilogram of gravel, and a jet airplane can combust fossil fuel extremely rapidly given its size. In addition, the services provided by entities do not necessarily increase in a simple way with mass, for example transportation can become less efficient due to increased traffic congestion caused by a greater mass of vehicles and roadways. Nonetheless, mass is a very straightforward starting point with which to understand the physical scale of the technosphere's main components.

## 4.1 Distribution of mass among MEUTEC categories

Despite their omnipresence in human lives, there has been relatively little prior work to assess the total masses of most technosphere components. Global studies of in-use material stocks have only been available for a little over a decade (Rauch, 2009; Müller et al., 2013; Glöser et al., 2013) and uncertainties frequently exceed a factor of 3, even for large aggregated categories (Lanau et al., 2019). In particular, the mass of materials in specialized buildings and heavy machinery is very poorly documented, and the masses of large public infrastructures such as dams and sewer systems are not typically available. Even for residential buildings, substantial disagreement exists in the literature. As a result, the estimates provided here should be seen as preliminary, and we hope that they will become better constrained through future work.

With those caveats in mind, we combine estimates from Wiedenhofer et al. (2024b), which are derived from the dynamic Material Inputs Stocks and Outputs version 2 (MISO2) model, with inventory-based estimates from Matitia (2022) and multiple other sources to quantify a subset of technosphere components. The methodology for combining estimates is described in Appendix A. Within each MEUTEC category, we differentiate buildings, total fixed stock and non-fixed (i.e. movable) stocks. Table A1 lists the estimated mass of each category, as a best guess, together with a representative uncertainty provided as a multiplicative range. For example, if the estimated value is 2 Gt and the uncertainty range is 3-fold, the actual value is very likely to lie between 0.67 and 6 Gt. The results are shown graphically in figure 3 for the total technosphere mass (including all of the built environment) and for movable objects on their own.

Despite the large uncertainties, it is quite clear that the largest two components of the technosphere, by mass, are the non-movable elements of the Ambient context and the Transportation system. Transportation is dominated by surfaces (37% of total, comprised of roads, railroads, bridges and tunnels), and also includes associated logistical infrastructure (poorly constrained). The inhabited environment is composed mostly of residential and service buildings (36 % of total mass), though infrastructure to control hydraulic flows (sewers, dams) also appear to be significant.

The movable parts of the technosphere account for only about 1.6 % of the total mass, $\approx$ 17 Gt. This is comparable to the total wet biomass of all animals on Earth ($\approx$ 20 Gt), or significantly more than the dry biomass of all animals ($\approx$ 4 Gt) (Bar-On et al., 2018). Terrestrial vehicles have the largest estimated mass of any movable category (3 Gt), though the within uncertainty range the terrestrial vehicle mass overlaps with the machinery and devices included in the technosphere manufacturing and construction categories ($\approx$2 Gt each). The mass estimates for Furnishings and Information storage, processing and display are somewhat smaller still ($\approx$1 Gt each), but again the uncertainties are very large and they overlap with a number of slightly smaller categories. Aircraft account for a remarkably small mass of $\approx$2 Mt.

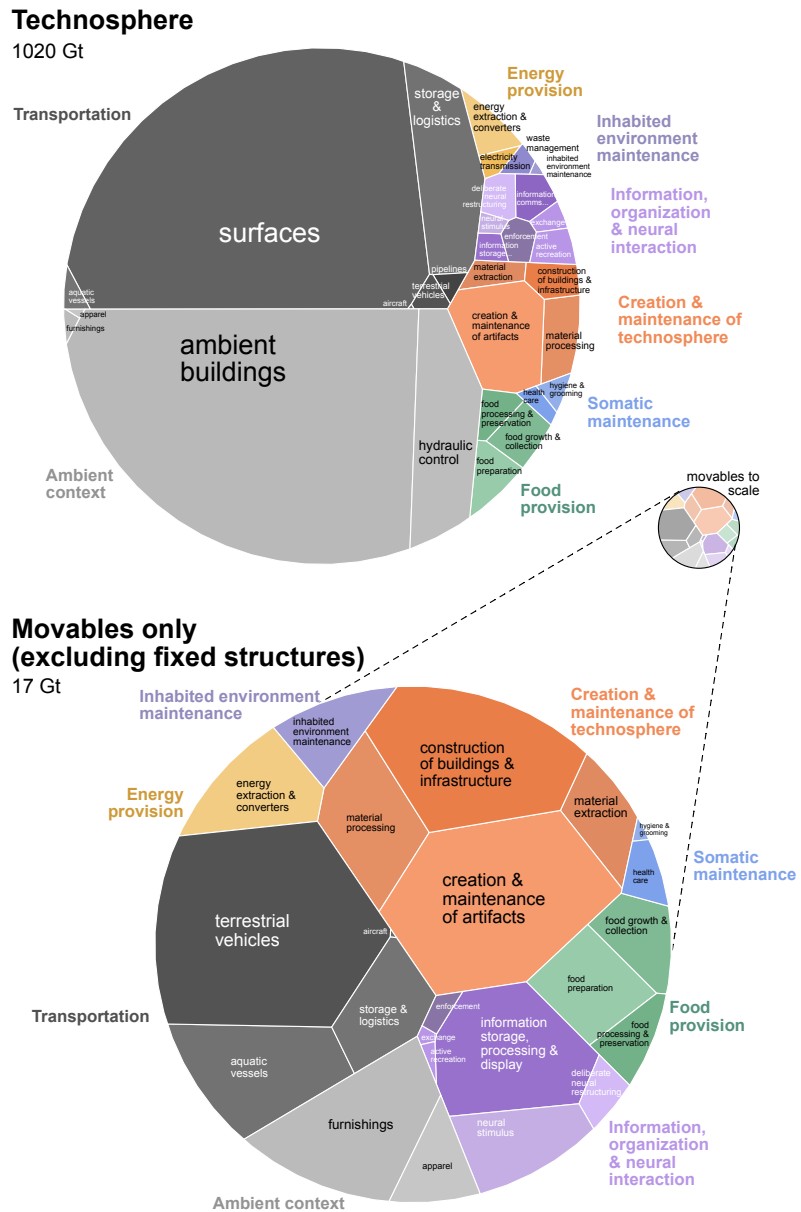

**Figure 3.** Approximate distribution of mass among the technosphere end-use categories. The area of each coloured region is proportional to the estimated mass. The top circle includes all fixed structures (buildings and infrastructure) as well as movable entities. The lower circle shows only movable entities, and the small inset (right side) shows the relative mass of movables compared to the whole technosphere (scaled by area). Because many categories include both fixed and movable components, they include significantly more mass in the full technosphere than in the corresponding movable category. The estimates include both inventory-based and material-inflow approaches over a range of years from 2015-2022, and the sums are therefore not representative of any single year. The estimated uncertainties for most categories exceed a factor of 3-fold due to lack of observational constraints. See Appendix A for estimation methodology.

## 4.2 Mapping the technosphere

We also provide an estimate of the overall spatial distribution of the two parts of the technosphere that comprise most of the mass: buildings, and the transportation system (Figure 4). Together these account for an estimated nine tenths of the technosphere. As detailed in Appendix B, the spatial distributions of these components are estimated from a combination of satellite observations and down-scaling of national data using surrogate local variables. The transportation system includes roads, railways, bridges and tunnels, passenger and commercial vehicles, rolling stock, commercial passenger aircraft, oil and gas pipelines, and the merchant fleet.

Although the total global masses of buildings and the transportation system are similar ($\approx 550$ Gt and $\approx 350$ Gt, respectively), they are distributed differently between world regions. The estimated transportation system mass is large relative to the building mass in Oceania and North America, and small compared to the building mass in Asia. As evident on the maps, the transportation system is particularly concentrated in central Europe, eastern North America, and eastern Asia.

It is important to avoid equating the geographic distribution of technosphere mass with direct Earth system impact. Technosphere-dense urban centres draw resources from the rural hinterland through processes that can cause dramatic changes (Brenner, 2014), even though the mass of technosphere in rural areas is relatively low. Nonetheless, the spatial distribution shown in Figure 4 provides a first-order picture of where the technosphere is most heavily concentrated, as a result of historical economic and social processes.

## 5 Dynamics of the technosphere

The technosphere is an extremely dynamic component of the Earth system, undergoing rapid internal transformations as well as driving large-scale changes in the rest of the Earth system such as climate change and habitat destruction. The technosphere is also a newcomer to the Earth system – arguably, the earliest components of the technosphere were stone and wooden tools, more than 2 million years ago (Otter, 2022), though these early hominin creations did not have the complex functional interconnections and quantitative significance that justify the term 'sphere' today. At that time, the global human population was likely only a few hundred thousand individuals, and the subsequent human population growth - over four orders of magnitude - has occurred symbiotically with the growth of the technosphere.

Here we briefly consider two aspects of technosphere dynamics. First, how parts of it accelerate the outcomes of specific human activities, and second, its growth over time.

### 5.1 Catalytic properties

We refer to the ability of a technosphere entity to accelerate the outcomes of an activity, for a given human time expenditure, as a *catalytic* property. Here we use the term catalysis in the chemical sense, since - like enzymes - these entities accelerate the creation of products without themselves being consumed. The idea of catalytic entities can be quantitatively expressed by the following differential equation,

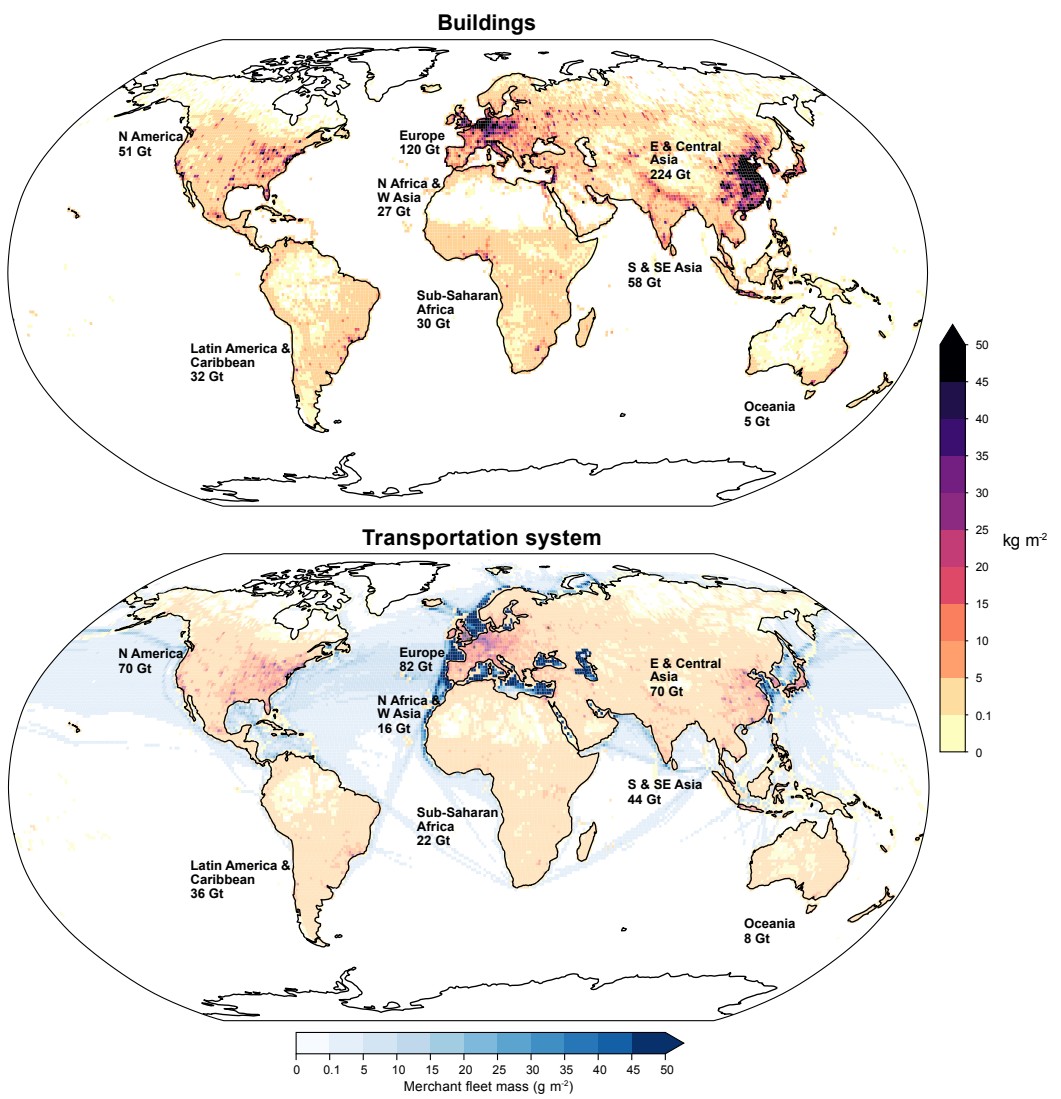

**Figure 4.** Spatial distribution of the two largest components of the technosphere by mass, at 1-degree resolution. Buildings (top) are taken from Haberl et al. (2025). Transportation system data (bottom) were compiled and distributed as described in Appendix B. The masses of ships used for marine transport are shown on a separate colour scale (bottom), whereas the same colour scale is used for all terrestrial masses of both panels (right hand side). Note that the palest yellow colour shows very low values, less than 0.1 $kg\,m^{-2}$, and a lower threshold of 0.001 $kg\,m^{-2}$ was used, below which the area is white.

$$\frac{dx}{dt} = A_x N T_x \epsilon_x \qquad (1)$$

Here, $x$ is an output (e.g. units of kg) produced at some rate over time $t$ (e.g. days) by the allocation of time to activity $A_x$ (e.g. hours per day) among a population of $N$ persons, $T_x$ is the mass of technosphere entities that play a catalytic role in the production of $x$ (e.g. kg), and $\epsilon_x$ captures all other factors involved in determining the overall efficiency of production (e.g. kg per person-hour per day per kg of $T_x$). Thus, an increase in the available technosphere entities $T_x$ will tend to accelerate the production of $x$ (increasing $dx/dt$) for a given amount of human time. We hasten to point out that $\epsilon_x$ is nonlinear, and caution against over-interpretation of this simplified formulation. For example it is well-known in economics that changes in $\epsilon_x$ tend to produce a saturating relationship with increasing investment in $A_x N$ (labour) or $T_x$ (productive capital) (e.g. Cobb and Douglas (1928)). Furthermore, catalytic processes can be complex, with a given entity playing catalytic roles in the production of multiple outputs. Nonetheless, this can serve as a starting point for quantitatively connecting population-level behaviour and the technosphere with changes in the human-Earth system.

## 5.2 Technosphere growth

The technosphere has grown particularly rapidly over the period for which ew-MFA estimates are available, which extends back to 1900 (Krausmann et al., 2017b; Wiedenhofer et al., 2019) as shown in Figure 5. The increase of technosphere mass since 1900 is well approximated by an exponential with a slope of $3.6\%\,y^{-1}$, equivalent to a doubling time of about 20 years. The details of the technosphere growth rates must be interpreted with caution, given the inherent uncertainties in the reconstructed masses based on ew-MFA. The material flows of major components are modeled from sparse data, and in-use lifetime assumptions are relatively simple. With those caveats in mind, it is notable that the exponential fit is particularly good since 1970, the period during which the data is likely to be most reliable compared to earlier time periods.

This rapid exponential growth can be attributed, at least in part, to the autocatalytic potential of the technosphere. *Autocatalysis* occurs when the products of a process increase the rate of the same process that produced them, thereby accelerating growth as the mass increases. The production of much of the technosphere, such as extractive and processing machinery and transportation infrastructure, clearly catalyze the activities of technosphere creation and maintenance, as discussed above. As a result, they can directly accelerate the overall growth, alongside other social and technical changes.

The autocatalytic potential of the technosphere sets it apart from the analogous creations of non-human organisms. Other organisms do modify their abiotic environments, including deliberate niche construction by animals. For example, birds build nests, beavers build dams and termites construct mounds. But although these modifications benefit their constructors, they do not catalyze their own further growth in the same way (Ellis, 2015). A termite mound does not directly contribute to the construction of further termite mounds other than by helping to ensure the survival of termites. As such, the masses of these other constructions are bound tightly to the masses of their creators. The technosphere, in contrast, has grown at a far higher rate than the human population, with the ratio of technosphere mass: human mass increasing by roughly a factor of 8 over the past century (from 18 t person$^{-1}$ to 140 t person$^{-1}$).

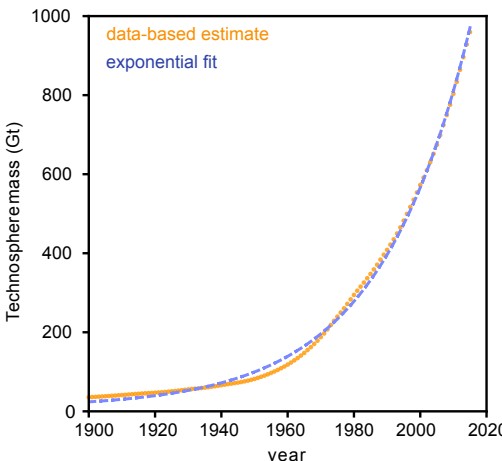

**Figure 5.** Growth of technosphere since 1900. Dots show estimated technosphere mass from Krausmann et al. 2018 with humans and domesticated animals removed. The blue line shows an exponential fit with growth coefficient $c = 0.036$, equivalent to a doubling time of roughly 20 years.

Autocatalytic growth of the technosphere at a given point in time can be described by a simple equation,

$$\frac{dT}{dt} = cT \tag{2}$$

where the rate of change of the technosphere mass ($T$, in $g$) is a linear function of itself. The coefficient $c$ (in $s^{-1}$) captures the degree to which the growth rate increases with the size of the technosphere. The value of $c$ would be expected to vary with many factors, including technological efficiency, societal organization, the labour pool, worker skill and resource availability,

so the causal role of the overall technosphere mass is hard to assess. As a result, the autocatalytic property, on its own, does not predict how the technosphere will grow in future, since $c$ can go up or down depending on a multiplicity of social processes. The value of $c$ is not an inherent property of the technosphere itself, nor does autocatalysis imply autonomy of the technosphere. However, if $c$ is constant over time, the relationship produces exponential growth of $T$.

The relatively good exponential fit of the technosphere growth since 1900 suggests that $c$ has indeed been relatively stable for

much of this time period, despite prominent historical events and human social dynamics that must have altered the value of $c$ to some degree (Krausmann et al., 2009; Wiedenhofer et al., 2013; Görg et al., 2020; Elhacham et al., 2020; Fischer-Kowalski et al., 2023). Past stability does not necessarily imply continued stability, and the value of $c$ could potentially decrease or increase in the future, depending on changes in human activities coordinated by social forces, as well as planetary feedbacks such as tipping points (Dietz et al., 2021; Wiedenhofer et al., 2024b). If $c$ were to drop to zero, the total mass would be

stabilized, while if it were to become negative, there would be a decrease of technosphere mass.

In addition, although we do not have detailed information on the size of the technosphere prior to the 20th century, simple calculations suggest that an exponential growth rate of 3% per year deviates strongly from its long-term average growth trajectory. For one thing, extrapolating this rate of growth backwards would imply that the technosphere would have had a mass

of 1 t around year 1330 AD, far lower than any conceivable value. For example, the single great pyramid of Giza, constructed in 2600 BC, alone weighs roughly 6 Mt. Assuming an total agrarian population in 2600 BC of 15 million (Fischer-Kowalski et al., 2014) with technosphere mass of 0.5-7 t per capita (Krausmann et al., 2016) implies a global technosphere on the order of 10-100 Mt, requiring an average long-run rate of growth <0.1% per year in order to arrive at the correct mass in 1900. Of course, one should not expect that this growth progressed continuously, given the numerous fluctuations of history. Many transitions between different socioecological regimes have occurred, such as the broad adoption of agricultural and increasing scale of societal organization, and – during the last few centuries – the agrarian-industrial transition, which is still ongoing (Fischer-Kowalski et al., 2014). The present global net growth rate of the in-use technosphere (110 Mt d$^{-1}$) - which amounts to roughly 20 great pyramids per day - obviously results from an acceleration that is anomalous in human history (McNeill, 2001). There are many societal innovations that could have contributed to the historical rise of the autocatalytic growth rate – coupled with severe sociopolitical ramifications, such as social revolutions (Lenton, 2016; Fischer-Kowalski et al., 2023). The greatly increased availability of technical energy due to the development of fossil fuels was likely a key factor within these innovations (Fischer-Kowalski and Haberl, 2007; Smil, 2007; Malm, 2013).

## 6  Conclusions

The technosphere concept provides a basis to think holistically about the vast physical construct that surrounds us. The definition proposed here is intended to be clear, distinct, and relatively unambiguous, in the hopes of better integrating this physical underpinning of human societies within the planetary perspective of Earth system science. Socio-ecological research provides a rich body of observations that can serve as a starting point for this integration (Haberl et al., 2019). Unlike most prior categorizations of the technosphere, which were based on material types or commercial features, the MEUTEC introduced here is based on the end-uses that motivate the creation of technosphere components, many of which align with human activities. The MEUTEC remains an imperfect categorization that could be improved through future work, and would benefit from comparison with alternative categorizations that capture other important features of the technosphere. Nonetheless, by grounding the categories on physically-oriented motivating end uses, we hope that this type of categorization can help to bridge the core features of economies and societies with Earth system processes.

As shown by the global data compilation, the mass of the technosphere is dominated by buildings used to provide a comfortable ambient context for humans, and by infrastructure and vehicles used to make the relocation of humans and materials faster and more convenient. The compilation shows that many categories are poorly constrained by data, a problem that is particularly pronounced for industrial buildings and fixed infrastructure (other than roads). Material Flow Accounting analyses have shown that the technosphere is composed almost entirely of geological materials: aggregate, brick, concrete, asphalt, plastic, glass and iron account for the vast majority (Krausmann et al., 2018). It is therefore predominantly a modification of lithospheric components. However, the technosphere has major impacts on the biosphere, accelerating the modification of the land surface and the extraction of organic matter from the biosphere, as well as on the atmosphere through the combustion of billions of tonnes of fossil fuels each year.

Our maps of the technosphere show the degree to which it is unevenly distributed over the Earth surface. The transportation system is particularly concentrated in Europe, eastern North America, and east Asia. Our appraisal of technosphere dynamics shows that the technosphere must have grown slowly over the Holocene, with average rates of less than 0.1% $y^{-1}$. This contrasts strongly with the past century, when growth rates exceeded 3% $y^{-1}$. Although many factors could have contributed to this acceleration, it appears likely that the strong autocatalytic character of the technosphere was implicated, linking it to other autocatalytic processes about which much has been learned. Importantly, autocatalysis does not imply autonomy – human engagement remains necessary for the creation and maintenance of the technosphere, and it follows that its future trajectory will be modified by societal processes, allowing the possibility of decreasing material throughput while providing high well-being to humans.

There remains much to be done to improve the understanding of the technosphere. For example, we have highlighted large uncertainties in the quantification of the fixed technosphere, primarily regarding industrial buildings and public infrastructure, which could potentially be greatly improved using remote sensing and machine learning. Further work could also elaborate details of the material composition and energy use of different technosphere components and link chemical elements within the technosphere with their Earth system sources and sinks, to build a unified understanding of how they contribute to global biogeochemical (or, perhaps, 'technogeochemical') cycles. There are many ways that the data compilation here could be used as a starting point for modeling aspects of the technosphere, potentially exploring links to time allocation, social decision-making or well-being implications. Coupled human-Earth system models incorporating a dynamical, fully integrated technosphere can help improve the understanding of physical constraints on system dynamics, supplementing Integrated Assessment Models to provide a complementary perspective on pathways towards long-term sustainability, as well as identifying potential tipping points. The trillion-tonne technosphere is a major component of the Earth system, and its evolution over the next century is likely to determine the future of climate – and life – for millennia to come.

*Data availability.* Global mass estimates derived from prior works are summarized in Table S4. The spatial building stock distribution can be downloaded from the DLR (German Aerospace) data product DLR EOC GeoService at https://geoservice.dlr.de/data-assets/h80jhtr41x48.html (DOI: 10.15489/h80jhtr41x48). The gridded data used for Figure 4 will be provided as part of the Surface Earth System Analysis and Modeling Environment (SESAME) Human-Earth Atlas Faisal et al. (2025) and as an electronic supplement on Zenodo.

## Appendix A: Estimation of technosphere composition by category

Constructing an estimate of the global technosphere composition by end-use remains highly challenging. Some categories are reasonably well-constrained by observations, but others are, at present, strongly limited by data availability. Our goal here is to provide an overview of the existing estimates as a starting point for future work, and to use them to provide current best estimates for all categories, which are necessarily highly uncertain. The available estimates are not all for the same year, but are generally for the period 2015-2021 unless otherwise noted. The sum of all individually-estimated categories, arrived at through

a combination of bottom-up and top-down approaches, is 1.03 Tt, equivalent to the total for year 2017 extrapolating from the Krausmann et al. (2018) material flow analysis.

Because buildings comprise a large part of the total mass and contribute to many MEUTEC categories, we discuss them first before proceeding to the other categories.

## A1    Buildings

We draw on three sources for global building mass.

Haberl et al. (2025) provide an estimate of building mass drawing on satellite observations of building volume (Esch et al.,
2022) to which they apply geographically-variable material intensities (i.e. masses of material per building volume). The total estimated stocks for year 2019 are 547 Gt (+/- 25%), of which 474 Gt are associated with residential use, 33 Gt with non-residential use, and 41 Gt associated with either residential or non-residential use. It should be noted that the identification of residential vs. non-residential buildings is methodologically challenging and should be seen as approximate.

Deetman et al. (2020) estimated building stocks based on a regression model of reported floor areas, interpolated across
regions, and simulated over time with a dynamic stock model. Their model differentiates residential from service buildings, but explicitly left out industrial and agricultural buildings, given the lack of statistical data on floor space.

Wiedenhofer et al. (2024b) use the MISO2 model to provide economy-wide, country-level estimates of material stocks across 13 end-uses. Two of these are buildings, divided between residential and non-residential, and suggest a total of 524 Gt in year 2021. Unfortunately, as for other estimates, the industrial, agricultural and other specialized non-residential building
masses are unconstrained.

We take Deetman's estimate for dominantly ambient environment service buildings (offices, retail and shops, hotels and restaurants) of 15 Gt, and assume the remaining 33 Gt of service buildings are more specialized to specific activities (e.g. schools, hospitals, public transportation, assembly buildings). Adding this ambient service building total to the residential building arrive at a total ambient building stock of 364 Gt, to which we attribute a factor of 1.5-fold uncertainty.

We aim for overall consistency between estimates by assuming the difference between the ambient building stock and the MISO2 total is accounted for by specialized non-ambient buildings, totalling 160 Gt, of which 33 Gt is non-industrial and non-agricultural. This suggests 127 Gt of industrial and agricultural buildings, roughly 1/4 of the total building stock. This is a highly uncertain value, and could be wrong by at least a factor of 2, which we hope can be addressed in future work. We then make a weakly-informed estimate of how this industrial building mass is distributed across the MEUTEC categories, to which
we assign a 3-fold uncertainty range. We caution that these fractions are very poorly constrained, and hope that they can be improved through further work.

## A2    Other categories

Because MISO2 provides a consistent, mass-balanced estimate that includes the entire technosphere, we use it as an overarching framework by relating the 11 non-building categories of MISO2 to the MEUTEC categories through a concordance matrix
(Supplementary Figure S4). Uncertainties tend to be large, with estimated ranges of 3 to 10. The largest uncertainty in terms of

total mass arises from the category 'Civil engineering except roads', due to its large mass (242 Gt) and diverse contents. These MISO2 estimates were supplemented additionally as follows.

For transportation surfaces, we supplemented the MISO2 estimate with the global estimate of 314 Gt in year 2021 from Wiedenhofer et al. (2024a) of all roads and railway infrastructure, including tunnels and bridges, constructed with archetypal material intensities applied to Open Street Maps data. We also used the similar estimate of 377 Gt provided by Matitia (2022) which also included CIA World Factbook road length estimates and applied slightly different material intensities. Averaging the three estimates suggests a mass of 375 Gt.

The energy provision category includes fossil fuel infrastructure as well as electricity production and distribution infrastructure. The masses of electrical transmission grids, distribution grids and transformers were taken as the median estimates of Kalt et al. (2021) for year 2017, totalling 1.7 Gt for electricity transmission with an uncertainty of roughly 50%. The energy extraction and conversion includes the power plant estimate of Kalt et al. of 8.4 Gt, including concrete in hydroelectric dams, and 0.7 Gt of fossil fuel extraction and refining infrastructure, of which 0.1 Gt is offshore oil platforms (Matitia, 2022). These estimates have significant uncertainty (range factor 4). The mass of pipelines was taken from Le Boulzec et al. (2022) as 3 Gt, which compares well to the independent estimate of Matitia (2022) of 1.8 Gt (uncertainty range factor 3).

We used bottom-up estimates from Matitia (2022) for agricultural tractors, passenger and commercial vehicles, rolling stock, the global merchant fleet, aircraft, military vehicles and weapons (see below for further details). The agricultural tractors and vehicles were interpolated to missing countries using a random forest model with GDP, total population, crop production, harvested area, percentage of urban population, year, and income class as predictors for machinery mass. We also used Matitia (2022) estimates for textiles, and the plastic components of furniture, electronics and health equipment as lower bounds on the corresponding categories.

**Appendix B: Estimating spatial distributions**

The spatial distribution of some technosphere components can be observed directly, such as roads (Wiedenhofer et al., 2024a). However, most technosphere mass estimates are only available on a jurisdictional basis, with a single value per country. To develop a harmonized, spatially-gridded raster dataset from jurisdiction-level data, our strategy is to employ the widely used dasymetric mapping downscaling method (Mennis, 2003; Faisal et al., 2025). The dasymetric method allocates data from jurisdictions to 1-degree grid-cells by using appropriate variables (referred to as surrogate variables). The jurisdictional data is distributed throughout the jurisdictional domain in proportion to the surrogate variable distribution. Thus, estimating the distribution for each category of technosphere mass requires an estimate of the value per country, and a surrogate variable to use for dasymetric redistribution.

**B1  Aircraft**

Commercial aircraft data were sourced from the Central Intelligence Agency (CIA) factbook. The average material composition of an aircraft was determined by taking the geometric mean of material intensities for five types of commercial aircraft as

| Level 1 | Level 2 | Movable | Total | Uncertainty |
|---|---|---|---|---|
| Creation & maintenance of technosphere | Material extraction | 0.4 | 6.8 | 5 |
| | Material processing | 0.8 | 14 | 5 |
| | Creation & maintenance of artifacts | 2 | 40 | 3 |
| | Construction & maintenance of buildings & infrastructure | 2 | 8 | 3 |
| Energy provisioning | Energy extraction and converters | 0.7 | 9 | 5 |
| | Electricity transmission | - | 3 | 5 |
| Transportation | Surfaces | - | 380 | 2 |
| | Storage and logistics | 1 | 57 | 3 |
| | Pipelines | - | 3 | 3 |
| | Terrestrial vehicles | 3 | 4 | 2 |
| | Aircraft | 0.002 | 0.002 | 3 |
| | Aquatic vessels | 0.6 | 0.6 | 2 |
| Food provisioning | Food growth & collection | 0.6 | 7 | 5 |
| | Food processing & preservation | 0.4 | 7 | 5 |
| | Food preparation | 0.3 | 10 | 3 |
| Somatic maintenance | Hygiene & grooming | 0.02 | 3 | 10 |
| | Health care | 0.2 | 3 | 5 |
| Inhabited environment | Inhabited environment maintenance | 0.4 | 0.4 | 3 |
| | Waste management | - | 3 | 5 |
| Information, organization & neural interaction | Information storage, processing & display | 1 | 4 | 5 |
| | Information communications infrastructure | - | 13 | 5 |
| | Enforcement | 0.06 | 3 | 10 |
| | Deliberate neural restructuring | 0.2 | 7 | 3 |
| | Neural stimulus & recreation | 0.6 | 2 | 3 |
| Ambient context | Ambient structures | - | 360 | 2 |
| | Furnishings | 1.3 | 1.3 | 5 |
| | Apparel | 0.4 | 0.4 | 5 |
| | Hydraulic control | - | 60 | 5 |

**Table A1.** Technosphere mass by category. All masses are given in Gt ($10^{15}$ g), for a movable 'non-fixed' portion, as well as for the total (including fixed structures). The uncertainty range is multiplicative factor. See Appendix for methodological details.

reported in Jemiolo (2015). The country level airplane mass data was proportionally distributed on airport counts per grid cell. The airport locations were obtained from http://ourairports.com. The ratio of plane capacities among these airport types is difficult to quantify, as it can vary greatly depending on a variety of factors such as the size and layout of the airport, the type and size of the aircraft it serves, and its operating procedures. We make a rough estimate based on general characteristics of these airport types, such that Seaplane Base : Small Airport : Medium Airport : Large Airport = 1 : 5 : 30 : 100.

## B2   Building material stock

The total building stock is taken from Haberl et al. (2025) and regridded to 1-degree resolution. Note that the mapped data are shown as previously published in Figure 4, which gives a slightly higher total (550 Gt) than the multi-source estimate shown in Supplementary Table S4 (520 Gt).

## B3   Merchant fleet

Country level merchant fleet data were obtained from United Nation Conference on Trade and Development (UNCTAD). Matitia (2022) utilized UNCTAD data to analyze per-country gross tonnage for five ship categories from 2011 to 2020, focusing on vessels with a gross tonnage of 11,000 tonnes and above. Steel mass per gross tonnage for these vessels was sourced from Kong et al. (2022). Because operational merchant ships are rarely in their home port, and are usually in transit, we dasymetrically mapped the global fleet mass using global shipping traffic density data.

## B4   Oil and gas pipelines

The spatial distributions of pipelines were collected from Sabbatino (2018). The oil and gas pipelines were converted from line to grids based on sum of pipeline length per pixel.

## B5   Roads and railways

The distribution of road and railway masses is taken from Wiedenhofer et al. 2024a and regridded to 1-degree resolution.

## B6   Rolling stock

The data for the number of registered locomotives, railcars, wagons, and train coaches were collected from a report of Union International des Chemins de Fer (UIC) and their data portal. The mass and material content of various rolling stock types were averaged from previously-published estimates (Delogu et al., 2017; Harvey, 2022; Kaewunruen and Rungskunroch, 2019). The country level rolling stock data was distributed proportionally to the railway densities, where the density data was collected from Global railways (WFP SDI-T - Logistics Database) at https://data.humdata.org/dataset/global-railways.

## B7 Terrestrial vehicles

The International Organization of Motor Vehicle Manufacturers (OICA) provided the information of registered passenger cars and commercial vehicles worldwide from 2005 to 2015. Passenger cars were categorized into large and small groups, with approximately 30.21% classified as "large" based on global new SUV registrations. Commercial vehicles encompass light commercial vehicles (LCVs), heavy trucks, buses, and coaches. It was assumed that the ratio of trailers to truck tractors is 1.5:1 globally, with estimates of 1.4:1 in Europe and 3:1 in North America (Matitia, 2022). We employed three distinct random forest

models for passenger vehicles, commercial vehicles, and trailers. These models aimed to estimate the number of vehicles per capita for countries and years where such data were missing. Predictor variables included GDP per capita, total road length, urban population percentage, and the year of analysis. All three models demonstrated a high level of accuracy, with test $r^2$ values exceeding 0.94, indicating robust predictive performance for vehicles across the specified categories. The country-level vehicle mass was distributed assuming that 5% of vehicles remained on the road, and thus distributed based on road density

(Meijer et al., 2018) while the remaining 95% of vehicle data was distributed based on population density.

*Author contributions.*  EDG: Conceptualization, Methodology, Formal analysis, Writing - original draft, Writing - review and editing, Visualization. AF: Investigation, Data curation, Writing - review and editing. TM: Methodology, Investigation, Data curation. WF: Methodology, Writing - review and editing. IH: Visualization, Writing - review and editing. HH: Writing - review and editing. FK: Data curation, Writing - review and editing. DW: Methodology, Data curation, Writing - review and editing.

*Competing interests.*  The authors declare no competing interests.

*Acknowledgements.*  We thank Ben Goldstein and Maxwell Kaye for insightful discussions, and Ron Milo and Daniel Horen Greenford for thoughtful feedback on the draft. This project was supported by the Canada Research Chairs Program fund number CRC-2020-00108 and the Natural Science and Engineering Research Council (NSERC) Discovery Grant RGPIN-2020-04889. HH and DW gratefuly acknowledge funding from the European Research Council (ERC) under the European Union's Horizon 2020 research and innovation programme

MAT_STOCKS, grant agreement No 741950, and Austrian Science Fund (FWF) project REMASS, doi:10.55776/EFP5.

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
