# Peer review of "Delineating the Technosphere: Definition, categorization and characteristics"

_EGUsphere, 2024_

## Author Response (AR1)

*We very much appreciate the insightful and challenging comments from all three reviewers. We have undertaken a major revision of the paper, including a rethinking of the categorization, and modifications of many of the figures and tables. The largest changes were made to the categorization section, expanding the connection to activities and including an entirely new figure to make the point. We also added detail and nuance to the descriptions of mass distribution and dynamics, and include new tables and analysis. We think the result is a great improvement on the original submission.*

*Below, please find a point-by-point response to all reviewer comments. In addition, a tracked change pdf shows the alterations to the text, though please note this does not include lists, tables or references (due to problems with latexdiff).*

RC 1

General comments

The authors provide a very valuable attempt at refining and constricting the definition and scope of the still somewhat loose concept of the ‚technosphere'. Through devising a classification scheme and appropriate metrics they aim to prepare it for integration into an Earth system analysis framework. This by itself is already applaudable as such an inclusion is still largely missing. In order to better delineate and understand the system-wide impacts of anthropogenic incursions into the ‚natural' Earth system metabolism and the dynamic coupling between the technosphere and the classical Earth spheres, the composition and quantification of technosphere entities, their respective properties and an overall growth function appear to be very meaningful for needed quantitative and qualitative assessments. As is obvious, and made also very clear in the paper itself, the value of the paper therefore lies not not so much in the originality of the approach but in the provision of such a conceptual and classificatory basis for singling out crucial features, attributes and evolutionary dynamics of technosphere components, as it is important for making the concept amenable for modeling the past and future (co-)evolution of the techno-Earth system. By doing so the paper largely follows a tradition in industrial ecology and socio-ecological metabolism research, with which the referee is only faintly acquainted and therefore can't properly judge the potential of the paper within that school of analysis. Instead the value of the paper is seen in providing, through the lens of and the expertise in material stock & flow analysis, a careful examination and thereby a helpful stimulus to advance the discussion about the mentioned practical ends for a joint assessment of technosphere and general Earth system dynamics.

*Many thanks for these supportive comments and excellent summary of the paper.*

Specific Comments

1) Classifying the entities of which the technosphere is comprised of is a bit like catching clouds with a butterfly net, or like an attempt to compress water in an air balloon: It will always escape and can never be arrested in a stable state. The approach the authors take is to guide their classification scheme via a number of defined end uses and derived general categories. In line with my general remarks above, this approach should certainly be granted publication, but the authors should be aware of – and possibly express in the paper – the taxonomic limitations of such an approach. A knife can be used to spread butter on bread, to kill a fish, deer or

human, to open a letter, to decorate a wall, to repair a gadget, and countless other ways. Similarly, material entities that fall under the rubric (in parlance of the authors) of food preparation could also be registered as sports equipment, furnishing, artifact creation and maintenance, domestic appliance, weapon, deliberate neural restructuring (through religious/ sacred objects), etc.

*We fully agree, and realized in retrospect that the original manuscript had come across as far too confident in the categorization. We have entirely rethought the categorization (now called the MEUTEC) and rewritten large parts of the manuscript in order to better express that this is just one, imperfect attempt, which will hopefully continue to be improved upon in future. (And thanks for the 'catching clouds with a butterfly net' phrase!)*

Also, the categories/terms offered in the paper are somewhat disputable. Aren't heating/ cooling devices more a means for survival in bitter cold or inhumane heat (or heat waves, hence a matter of public health) than just providing "somatic comfort"? The authors briefly mention physiological limits (line 192) but the Level 2 designation of the Ambient Context primary category in Table 2 and general definition in Figure A1 do not incorporate such considerations of fundamental life support systems and thereby seem ill-named and/or ill-defined.

*We see why this could have been confusing. In the revised manuscript there is no longer an Ambient Context primary category.*

A further example is the designation of "Non-informational equipment and buildings specialized for teaching, religious activities and research" in Table 2. It seems unclear why a white board or a bible is non-informational and would not fall into the Information category as material media that communicate, store, and process information. Equally ambiguous is the filing of buildings into different categories. Each architectural structure on this planet is a complex assemblage of multiple functions or end uses, houses a myriad of different technosphere entities, is created out of myriads of different lithospheric materials, processes a variety of energy conversions. Thereby a classification into residential buildings (Ambient Context) and buildings for growing food or extracting materials (Activity-specific) seems quite arbitrary.

*We agree regarding all of these challenges, which we have tried to address by two complementary strategies. First, the overhaul of the categorization removed the primary distinction between 'Activity-specific' and 'Ambient context', going now straight to the more immediate descriptions instead. Second, we have tried to improve the discussion of how entities are categorized, as well as providing a more detailed table (S1) including priority schemes to navigate ambiguous boundaries. The result is still not perfect, but we think it is a big improvement and are grateful to the reviewer for these comments.*

Moreover, the technosphere mutates over time, and to some extent also end uses mutate. Although the paper acknowledges, and even attempts to estimate, the specific dynamics of technospheric growth over time, its categorization effort cannot otherwise than result in somewhat artificially arresting technospheric entities in terms of contemporary end uses which might be outdated very soon and might have not existed in previous periods of technosphere evolution. As the cited Chris Otter expresses: „Homo sapiens and the technosphere coevolved", yet such a coevolution can't be captured with a static and fixed classification entirely focused on the here and now (especially if it something superhuman as the technosphere).

*Thanks for these comments, which inspired the addition of a new section discussing hunter-gatherer material culture in contrast to a modern goods classification (the CPC). As shown in*

*the revised Table 2 and Supplementary Table 2, the entities found in both of these very different contexts can be readily classified with the MEUTEC, yet their comparison also clearly shows the fact that new aspects of the technosphere can develop over time. We think this illustrates the point of the reviewer (and of Chris Otter) very well.*

In sum: A classification of the diversity, variety and variability of the technosphere assemblage cannot do other than amounting to an exercise in futility. This is not the authors fault; such ambiguities are fundamental and likely can't be resolved to general satisfaction. However, attempts such as the one made in the paper can still help to refine our understanding of the technosphere composition and shed light on its main drivers. The referee only thinks that such more modest and limited aims should be addressed more openly in the paper. Just in general, the paper would profit from a bit more reflexive stance on the approach chosen. The made claim that "we are now in a position to characterize the basic features of the basic attributes of the technosphere in a holistic, exhaustive and exclusive manner," (line 223) is hardly defendable.

*We see the point but would suggest that, rather than an exercise in futility, it is a useful exercise that simply does not have a unique solution. We very much appreciated the suggestion to shift the tone, and hope the revised text is appropriately reflexive.*

2) The focus on mass analysis of technospheric components (and not, for instance, structural function) necessarily leads to rather modest gains in knowledge, however complex and challenging such an assessment already is. This was also the problem of the cited Zalasiewicz et al. 2017 paper, although the authors of the present paper address and resolve some of the issues that came with it, e.g. by excluding human-modified soils.

*We entirely agree that mass, on its own, is insufficient for a thorough understanding the technosphere. We will clarify that the mass distribution shown here is intended to provide an overview of this classification and framework, and that many interesting aspects remain to be resolved next, such as energy conversion rates, metal composition, toxin production, lifetimes, etc.*

An example here is the class of the Informational, a category extremely important for the (autocatalytic) growth of the technosphere (e.g. Felix Creutzig et al. 2022). As the authors rightly mention in line 244 "The Informational category, despite its prominence in human affairs and experience, accounts for only a small portion of the total mass". That by itself should be a warning sign for the authors. Weight clearly doesn't equal importance. It doesn't tell you what drives, accelerates or decelerates the technosphere's overall power to establish itself as a new Earth sphere. It tell's you how much heavy material is now installed (and in use) on Earth's surface, or better: how much material is converted from the lithosphere to the technosphere, but it does not assess – or "weight", for that matter – the structural, functional and dynamical nature of the latter.

*Thanks for the suggestion to look at Creutzig, 2022, which we found a very helpful reference and have now incorporated into the discussion of informational entities.*

Another example is the discussion of the geographical distribution in the mapping subsection 4.1.2 and Fig. 3 which lacks not only insight but sophistication. As Henri Lefebvre has already shown in the 1970s, the planet is already largely 'urbanized' as it is a necessary background resource for sustaining what once was called cities or agglomerations. Again, a focus on simple mass accumulation does not help to explain how agglomerations and the planetary "hinterland" crucially interact to form a planetary-scale agricultural/extractive/consumptive anthrome (see e.g. Neil Brenner, 2014).

*Good point, we have added this to the discussion of the maps.*

3) The paragraph discussing the choice of focusing the analysis on components 'in use' (starting at line 98) might merit from briefly addressing the concept of the 'technofossil' as a sort of counterpart (e.g. Zalasiewicz et al. 2014, or https://www.anthropocene-curriculum.org/anthropogenic-markers/novel-materials-and-technofossils/contribution/the-technofossil-record-where-archaeology-and-paleontology-meet).

*Great suggestion! Added, thanks.*

Technical comments

- The paper misses consistency in terminology used across tables, text and figures. These are essential, however, for the reader to decode, for instance, Figure 2, and align it with what is listed in Table 2 and Figure A1.

*Apologies for these oversights in the original submission. We've now changed all the terminology again, but hopefully it's consistent this time.*

- Line 110: correct spelling is foraminifera

*changed*

- The sentence starting in line 219 should be moved up to the opening paragraph of subsection 3.1. Thereby the reader is better prepared to what is to come and is helped to navigate the categories.

*revised*

- Suddenly, in the last sentence (line 334), the "trillion-ton technosphere" pops up, probably a remnant of the earlier analysis by Zalasiewicz et al. That number doesn't appear earlier in the paper, if I am correct.

*The total mass estimates for years 2015-2020 come out very close to 1,000 Gt, or 1-trillion tons, which may not have been obvious in the original tables. This now appears a few times in the revised paper.*

RC 2

The paper aims to refine the definition of the technosphere, introduce the End-Use Technosphere Classification (EUTEC) for systematic categorization, quantitatively assess the mass and spatial distribution of the technosphere, and understand its dynamic growth patterns. In addition, it seeks to establish an interdisciplinary foundation linking the material composition of the technosphere to its functionality and impact on human well-being, while emphasizing the need for further integration with Earth system science. I think these are all important and necessary goals, and the authors make substantial progress toward some of them. However, I am very ambivalent about the manuscript as a whole. I very much appreciate

the enormous effort the authors have made to quantify all aspects of the technosphere. This is already an important and substantial contribution, despite the fact that there is a lot of uncertainty in some aspects of this quantification.

It is the analytical part that I struggle with. I have some disagreements with some of the definitions of the technosphere and the proposed dynamics of the technosphere as described by the authors. I also do not see how the authors suggest that the analytical part contributes to the inclusion of the technosphere in Earth system models.

My main issues concern: (1) the suggested autocatalytic nature of the technosphere and the implications of these assumptions for the potential modeling of the technosphere in an Earth system model, (2) the boundary definitions of the technosphere,  (3) the overemphasis on infrastructure growth to satisfy end-user demand, to the exclusion of all other factors that shape the technosphere, and (4) the suggested intrinsic dynamic of the technosphere as described by equation 2.

In addition, I agree with the first reviewer on all their major points so will not reiterate them here.

(1) I would disagree that the technosphere as defined here (i.e., excluding „human activities or mental constructs such as institutions, corporations, or social norms") can usefully be described as autocatalytic. I agree that the quality and quantity of artifacts can determine the quality and quantity of artifacts that can be produced, but I would argue that the technosphere as defined here does not possess the inherent properties required for self-sustaining, self-enhancing growth and evolution. Instead, it is mostly shaped and driven by external societal and human factors (besides the availability of material inputs).

The development and evolution of technological systems is highly dependent on human innovation, decision-making, and societal needs, and is driven by societal goals, economic factors, regulatory environments, and cultural values (all of these factors often shaped by whoever holds power in these societies). Without the input of human creativity and direction, technological systems do not inherently create or improve themselves (the current AI hype notwithstanding).

In my understanding, for a system to be autocatalytic, it must have self-sustaining mechanisms that promote its own growth and complexity. In the absence of human and societal inputs, the technosphere (as defined here) lacks the intrinsic capacity to innovate or evolve autonomously. Technological systems require maintenance, updates, and guidance from human agents. While there are certainly feedback loops within technological systems (e.g., improved machinery leading to more efficient production of new machinery), these loops are initiated and maintained by human intervention and societal demands.

I admit that some of my disagreements may be more a matter of semantics. For example, I do not find the arguments in Herrmann-Pillath 2018 based on Stuart Kauffman's notion of autonomous agents very convincing, nor am I sure that the definitions of the boundaries of the technosphere used there and here are consistent. If one understands industrial metabolism as a property of industrial societies rather than a property of industrial systems (as originally defined by Aryres), I would agree with the characterization of this coupled system as autocatalytic, where flows of energy and material are mediated by both physical infrastructures, symbolic structures, and human labor. Of course, I could very well be wrong about this, or misunderstand the intent of why it is appropriate to call the technosphere as

defined in the manuscript autocatalytic. My main concern, and the reason I am spending so much time on this issue, is what follows from this assertion. I will return to this below.

*We appreciate these points, and entirely agree that the technosphere is not self-sustaining, but is entirely shaped and driven by human factors: it is not, by any means, autonomous. Strictly, we had been thinking of the term in the sense of the chemistry definition: a reaction is autocatalytic if one or more products of the reaction accelerate the reaction rate. Because exponential growth is a feature of autocatalysis, we find it a useful description.*

*The revised manuscript includes substantial discussion of the 'catalytic' nature of many technosphere entities, defining this term in the chemical sense, and explicitly linking them to human activities with an equation. We use this as a conceptual improvement on entities that were originally termed 'activity-engaged'. In this way, when 'autocatalytic' comes up later, we hope that it will be abundantly clear that humans are implicitly involved in the autocatalysis, as suggested by the reviewer (though we still keep the technosphere and humans separate).*

*We also added a discussion to clarify that autocatalysis does not mean autonomy.*

(2) My next issue is with the third definition paragraph of the technosphere regarding when an „in-use component ceases to be fit to serve an intended end-use". You concede that this „boundary is often subject to social characteristics" but then you claim that it „is nonetheless observationally quite easy to identify." I am not convinced that it is so easy, and I am not sure how significant the term "intended" is in this statement. As such, I find this argument perhaps overly simplistic and potentially biased toward capitalist or Western perspectives that do not take into account the nuanced and subjective nature of determining when an object is no longer fit for its intended end use. For example, I live next to a dilapidated building that is being used as an impromptu skate park and a place for people to take drugs. It is not being maintained in any relevant sense, but I would still consider it part of the technosphere. Or, for that matter, the Pyramids of Giza probably left the technosphere by your definition before they were repurposed as a tourist attraction. I would argue that many structures and objects retain their place in the technosphere even though they no longer serve their original or intended purpose. I think the definition overlooks the different ways that different societies might use and maintain objects, reflecting a bias that doesn't apply universally to all cultural contexts. The reason you give for excluding waste is that it is difficult to determine when waste ceases to be waste. I would argue that for many parts of the technosphere it is difficult to judge when they begin to be waste, so we are simply shifting the conceptual uncertainty from one place to another.

*Many thanks for this provocative comment, we spent a good deal of time thinking about the dilapidated building next door. We have removed the 'observationally easy to identify' and revised the discussion of the waste boundary (including, instead, a medieval fortress as an example) and hope the reviewer will find that it is now better explained.*

(3) I also struggle with the justification for the end-use categorization, which says that "human end-use outcomes are what motivates [sic] the existence of the technosphere - every component of the technosphere was motivated by at least one type of intended end use". For example, you say that buildings exist or are constructed primarily to "provide humans with a more comfortable, attractive, or otherwise desirable immediate environment". There is, of course, some truth to this (while acknowledging the first reviewer's criticism that this is often more a matter of survival than comfort and beauty). However, much of the construction of buildings is actually driven by capital looking for investment opportunities. This can be seen everywhere, perhaps most dramatically in the infamous Chinese "ghost towns", where the end

use of providing shelter for people is not even a secondary use, as it would devalue the investment. The same could be said for most types of infrastructure. Many additional lanes for roads are not built to make mobility more efficient and comfortable, as decades of transportation studies show that building additional lanes does not reduce congestion or make mobility more effective. So I, for one, would strongly disagree with the conclusion (l320) that the infrastructure that locks in car-dependency in may cities exists primarily for the purpose of making the „relocation of humans and materials faster and more convenient." Or, for example, Timothy Mitchell (Carbon Democracy) has written extensively on what motivated and shaped the development of the global oil infrastructure, which was only tangentially related to an existing demand for the end use of oil.

*Thanks for these thoughtful points, which have led to additional principles in a new section on 'Assigning entities to categories' which hopefully better clarify what is meant by an 'ostensible end-use' as opposed to social, economic or cultural significance. We also mentioned the fact the services provided by the technosphere do not necessarily increase with the mass of the corresponding entities. We recognize that this cannot be a perfect system of categorization, as discussed in response to Reviewer 1, and hope that it can be further improved in future.*

More generally, I think there is much to be said for the argument that infrastructure can very often be seen as a physical manifestation of power rather than simply a neutral response to a demanded end use. Political economy or science and technology studies would argue that technological systems and infrastructures are not neutral, but are imbued with the values and interests of those who design, build and control them. Infrastructure can thus be seen as a materialization of social order and power dynamics that shape how people interact with technology and each other (e.g. works by Foucault, Harvey, Castells, Levebre, and many others). This reference may also be helpful in the context of this manuscript [1]. This concern also comes back to the characterization of the technosphere as mainly autocatalytic, which seems to negate or at least disregard these concerns. I think it is not only wrong, but even dangerous, to ascribe a largely intrinsic and autonomous dynamic to the technosphere and to model it in an Earth system model in order to explore possible future trajectories. To be somewhat polemical, there is a danger of cementing a destructive status quo, based in no small part on a particular (symbolic!) ideology, and recasting it as a neutral and intrinsic property of inanimate matter.

*This point is well taken. We have not gone so far as to include discussion of political economy on the issue, as we feel we would be out of our depth. However we have tried to reword any segments of the manuscript that seemed to give the impression that we were implying a neutral and intrinsic property of inanimate matter, and - as mentioned above - have clarified that autocatalysis is by no means indicative of autonomy.*

(4) The whole focus of my critique on the autocatalytic (and thus autonomous) nature of the technosphere is driven by my concern about what the implied end use of equation (2) is. This is not clearly stated. However, you state that a major goal is to integrate the technosphere into Earth system models. There are few specifics on how to do this, but since you give this equation, I assume you are suggesting that it would be helpful. Equation 2 fits one or more coefficients to the historical growth function of the technosphere. This describes the empirical observations, but it has no explanatory power and is therefore not really suitable for predicting much about the future. On the one hand, we can already see a certain saturation in the throughput of non-metallic minerals in developed economies, and the exponential growth is coming from regions that are currently urbanizing/industrializing and thus accumulating large stocks. Also, similar levels of end-use in the US and Europe in terms of roads and buildings are achieved at very different levels of material use, and it is unlikely that Europe will grow these stocks to US levels.

Overall, since any kind of "sustainable" social metabolism is likely to require a reduction in material throughput, which is quite achievable in rich societies while increasing well-being, I do not think it makes sense to extend Earth system models with an "autonomous" technosphere module that exhibits intrinsic exponential growth.

*We agree that building an Earth system models with an autonomous technosphere would be a bad idea! Again, we apologize for not clarifying better that autocatalytic does not equal autonomy, and hope the reviewer will find that the revised manuscript does a better job at showing the central role of human agency, through what is now a much more prominent discussion of human activity. We thank the reviewer for showing how important this is.*

Small things:

Table 2: why only fluids in pipelines and not also gas?

*clarified with examples*

Figure A1: The category "Technosphere" should be "Technosphere Creation & Maintenance" to avoid throwing the reader into a recursive loop. =)

*recursion eliminated*

On the one hand, I appreciate the motivation and goals of the paper, and I think that the empirical part of the paper should definitely be published because it is a valuable contribution to knowledge. Here I would urge you to consider publishing the data separately in a data repository (like zenodo) and ideally also any software code used to transform the already published data into your results.

*Thanks, we will make the data available.*

As may have become clear, I am less comfortable with the analytical part. My personal suggestion would be to simply remove that part altogether. I realize that this is probably not a suggestion you will want to follow. In that case, I think a fairly major revision of this part is needed, one that seriously addresses the issues I have raised.

*The reviewer's intuition was correct - major revision it is!*

The questions I hope to have answered are:

Why does it matter whether the technosphere is autocatalytic or not?

*In our estimation, it is a useful property of the system to be aware of, just as it's useful to be aware of for other autocatalytic systems. However, its predictive power is limited.*

 What implications does this have for how the technosphere would be implemented in Earth system models?

*The technosphere could be implemented in many ways, of varying complexity, and the present authors do not pretend to know what all of these ways might be. However it is our understanding that the autocatalytic property of the technosphere is real and important, and therefore ought to be a feature of a technosphere model (though it could be an emergent outcome, rather than an a priori assumption).*

How is such a characterization justified or operationalizable if all symbolic structures are excluded?

*The role of symbolic structures is unquestionably of great importance - we would see these as altering human activities, in a simple sense, though we do not go into that in the current work.*

Why the focus on end use?

*We have added much more discussion of human activities and why these can be usefully coupled to end uses to provide functional insights.*

How should dynamics of the technosphere that are not driven by end-use demand be included in a possible operationalization in Earth system models? What about the role of power shaping the trajectory of the technosphere (either political or economic)?

*These are great questions, which we do not feel can be answered in the current manuscript. We have mentioned these as a future direction in the last paragraph of the conclusion.*

How is this definition and categorization of end use likely to hold in future societies that may have very different social and economic relationships with their material environment?

*Great question! We have added a new discussion of this that contrasts hunter-gatherer technology with modern goods, 'Application to existing lists of entities'. Thanks for the suggestion.*

What is the role of Equation 2 with respect to the inclusion of the technosphere in Earth system models, and how is an intrinsic exponential growth dynamic useful for modeling potential post-growth, steady state, or other economic imaginaries where infrastructure stocks are saturated?

*We have added a paragraph discussing exactly this, referring to how changes in the coefficient 'c' capture the range from growth to stability to collapse.*

I apologize for the length of this review and hope that you will find some of the points I have raised interesting and helpful in improving the manuscript.

*On the contrary, many thanks for the deep engagement! We think the revisions arising from the comments have significantly improved the manuscript and hope the reviewer will agree.*

RC 3

General comments

The authors attempt to define and quantify the technosphere, to assist its integration into Earth system analysis. Their approach differs from prior work largely by centering their definition on human-intended end uses of technosphere elements. This is an ambitious, challenging and, in my view, important task. I think the authors' approach has many strengths and is a substantial step towards their stated goals. However, there remains work to do before they can defend their claim that they are able to exhaustively and exclusively characterise the technosphere. Like the other reviewers I support publication of at least part of the text, on the condition that

the authors revise it to be more explicit about the limitations of their taxonomy and more clearly identify the areas where more nuanced development is needed.

*Many thanks for the supportive comments, we have indeed undertaken a major revision including much more explicit discussion of limitations and nuance.*

Specific comments

Reviewers 1 and 2 raise many interesting lines of commentary that I think stand well on their own. I would like to extend one line that runs through both of their reviews: that, however convenient it may be for accounting purposes, identifying the technosphere with material composition, detached from the cognition within the social processes that coevolve with the technosphere, does not achieve the authors' goal of exhaustive characterisation. For want of a better analogy, this seems like trying to describe how and why a computer is useful for its user by describing its material composition while ignoring the software installed. I agree with reviewer 2, that to understand the autocatalytic behaviour described by the authors, one needs to consider the sociotechnosphere (by which I mean something like 'technosphere as defined by the authors plus social cognition'), not just the technosphere as defined. I also agree with reviewer 2 that much of the technosphere is better understood as resulting from emergent socioeconomic dynamics that can be quite disconnected from the needs and intentions of the individual humans who interact with the resulting technosphere elements.

*We absolutely agree that the coupling of society and technosphere is an essential part of the dynamics. In the revision, we have added a discussion of how technosphere entities catalyze the outcomes of activities, which we hope helps to explicitly show the integration of humans in technosphere creation and maintenance. We have also added to the discussion of autocatalysis multiple mentions of the societal role. However we would resist lumping society and the technosphere together as a 'sociotechnosphere', as this feels like it would become something more vague and inherently unphysical.*

However, I part ways somewhat with reviewer 2 on the value of the manuscript's analytical section, again with the proviso that equation 2 describes the sociotechnosphere and not just the technosphere. The point here is not to include the biomass of humans but the cognitive processes that facilitate the autocatalytic dynamic. In my view, that the data demonstrate exponential growth of the mass of the sociotechnosphere does have explanatory power insofar as it places that system within a broader class of exponentially growing autocatalytic systems about which a lot has been learned. Although we may like to believe that by exercising our collective agency we can steer the system and convert exponential to logistic growth on the relevant timescale, it may be that the emergent system dynamics are now beyond our control and overshoot-and-collapse is inevitable. In my opinion, presenting and interpreting the century-worth of data of the system's 'gravimetric growth' is very worthwhile. To agree again with reviewer 2 though, I suggest presenting equation 2 together with a discussion of its potential limitations; future data may reveal a need to elaborate the model to accommodate saturation, collapse or other potential divergences from exponential growth.

*Thanks for the support of the autocatalytic discussion, which prompted us to keep it. We have added the discussion as suggested.*

I would also like to venture in a direction not explored by reviewers 1 or 2, but still on the theme of limitations due to grounding the taxonomy purely in material composition. I offer a few rather pedantic 'thought experiments' that are intended to help the authors stress test their definitions by considering edge cases. I agree with reviewer 1 that aiming for a perfect taxonomy is a fool's errand, so I'm not suggesting that the authors need to address everything I raise here in

detail before publication. Rather, I suggest considering whether they can use my provocations to strengthen their taxonomy by refining definitions.

In short, the weakness I see is that where something fits in the authors' taxonomy depends strongly on scale of analysis and context. From line 70, the authors state that they 'follow the Earth system science convention of defining a sphere in terms of the nature of the matter of which is it comprised, rather than in terms of specific processes or fluxes.' They illustrate this with the example of the atmosphere, which is defined by its location above the Earth's surface and its chemical composition rather than its internal dynamics. This seems to imply a kind of coarse-graining over the atmosphere at some large scale of analysis, such that only its average material composition over that scale is considered.

This approach may run into trouble when trying to identify boundaries between Earth system spheres, as the authors do later. If a thing is categorised according to its average material composition, why exclude a recently dead organism from the biosphere? If it decomposes through heterotrophic respiration, presumably the mass assimilated by the heterotrophs is still part of the biosphere. I would agree that the original organism is no longer part of the biosphere but if this is true while some of its material still is part of the biosphere, then the organism was categorised not by its material composition but rather by its functional organisation (the thing that make it alive).

*Thanks for the helpful challenges. We have revised the text in this passage to clarify that heterotrophic assimilation does indeed maintain mass in the biosphere, and that the functional organization is a key part of the matter (not just the chemical composition).*

Presumably, individual metabolite molecules within the organism are not alive but are categorised in the biosphere because they are located inside a living organism. If this is the case, why isn't a person's pacemaker part of the biosphere? If the answer is that its average material composition differs from the rest of the host's body, would the distinction still hold if it were possible to build a pacemaker from biomaterials?

*We added a specification that living things are composed of cells with active ribosomes, since this seemed like a universal and characteristic feature of life. If biomaterials of the future are composed of cells with active ribosomes… well… the categorization may need to be revisited.*

By analogy, why isn't a person inside a building a part of the technosphere? If they are, is their biomaterial-based pacemaker now also part of the technosphere? If they step outside, does it become part of the biosphere? Conversely, given that human settlements are autocatalytic dissipative structures like organisms, and resulted from the same evolutionary process, are they really 'nonliving matter'? This seems clear at the scale of a brick but is perhaps less clear at the scale of a city. If a genetically modified sheep is part of the biosphere, is its DNA, which is no more alive than the brick, part of the biosphere or technosphere? In which sphere would a genetically modified virus be categorised (given that the living/nonliving status of viruses is controversial even without engineering)? What about a completely novel, artificial virus? Would the answer change if it were inside/outside an organism? Do these examples demonstrate that applying the authors' taxonomy unambiguously may not be possible without first defining a scale of analysis and considering context? If the authors wish to connect their definitions to accounting systems that have different resolutions, these issues may become crucial. Perhaps the authors have faced such questions before in their work on sociometabolic accounting (contrasting MEFA with MuSIASEM, for example). I think they should make some effort to address these issues in their paper and justify any relevant simplifying assumptions underlying their taxonomy.

*Great points - we have added a section on 'assigning entities to categories' where we attempt to clarify these thorny issues, including a definition of 'entity', and hope the reviewer will find it an improvement.*

94-97: The issue of human agency arises here and while this seems to support my earlier comments about the necessity of including (social) cognition in an adequate definition of the technosphere, I would like to briefly also consider a less anthropocentric view. The existence of genuine agency continues to be controversial even in humans, and the existence of technologies with behaviour that is indistinguishable from human agency is becoming ever more plausible. Accordingly, technologies that make their own institutions and norms within systems of collective computation are also becoming increasingly plausible. Might it therefore be too restrictive to exclude phenomena like institutions and norms from the technosphere even if the human versions are excluded? Even in the absence of strong AI, some form of artificial life may be achieved. Would this be part of the technosphere or biosphere? Perhaps the taxonomy needs a tiebreaking rule for such cases?

*Fascinating question - we have decided to punt on this one and just refer to 'social constructs'! We added text to justify why these are inherently biological (couched in neurons).*

Technical corrections

- At first glance I found the title a bit confusing because the word 'resolving' is ambiguous (Is the paper trying to fix the technosphere?). The abstract clarifies the meaning but the title alone does not clearly communicate what the paper is about.

*Thanks for raising this potential confusion, we have changed the title to 'Delineating the technosphere: definition, categorization and characteristics'.*

- In Table 1 caption, is 'of which they are comprised' correct English? Presumably the Earth spheres comprise various kinds of matter rather than the other way around. Maybe use 'of which they are composed'?

*Thanks! Fixed*

- 33: It's unclear whether the 20 TW refers to the technosphere or terrestrial above-ground NPP.

*Fixed*

- 70-71, and also 178: Unusual use of 'comprised' again.

*Fixed*

- 259: Three million years ago seems to be too early for reference to a human population. Hominin population might be more accurate.

*Fixed*

- 263-265: Before and after equation (1), you carefully specify units in which all the terms are measured. Shouldn't the equation be insensitive to unit selection, provided all terms are consistent? I note that you specify kg for some terms and g for others; is this intentional?

*Fixed*

---

## Author Response (AR2)

March 21, 2025

Dear Dr. Donges,

Many thanks for obtaining the second round of reviews on our manuscript. As the second reviewer had no further comments, we include below only the comments from the first reviewer, followed by our response (in blue).

I appreciate the time and effort you took to revise the manuscript in response to the three reviewers' feedback. Some changes were, in my view, very successful in addressing the concerns raised. I will admit that in other areas where you have sharpened your arguments, my disagreement has actually deepened.

Without going into much detail, this concerns mainly your arguments on how to disentangle social from physical processes to delineate the boundary and dynamics of the technosphere (and still its autocatalytic nature), and how you justify the end-use categories. I do not have the bandwidth right now to go into a detailed discussion, and I think it would also not be productive in changing your collective views on these issues.

That said, for the first issue, I do not think that it is currently possible for either side to argue our position from first principles, so there remains an element of judgment. The second issue is also one of judgment: whether or not the usefulness of the end-use categories justifies their shortcomings. It is your paper, not mine, so I think it is fair to concede these unresolvable points to you. The manuscript is now (somewhat) more reflexive, so it should be clear to readers that this is a contribution to efforts to "delineate the technosphere" and not the definitive answer in all regards. The manuscript is well written, and your positions are now argued clearly enough that future research can agree or disagree with them. Given the complexity and scope of the paper, I think this is very laudable and well justifies the publication of the manuscript in its present form.

I was going to leave it at that, but there is one thing that I cannot leave uncommented. Particularly considering the involvement of three preeminent scholars in social metabolism research from ISE/BOKU, I was reading the paragraph starting in line 424 in open-mouthed disbelief.

We were disappointed to hear that the end-use categorization remains unsatisfying, but also recognize that there can be no universal solution, and agree with the reviewer that multiple approaches will ultimately be helpful. Nonetheless we took this opportunity to attempt an improvement of the categorization and its description, and have made one last revision of the text to strengthen the reflexivity and further clarify the conceptual basis.

As part of this we shifted the language away from activity-centered to motivation-centered, as we feel this is easier to understand. We included text to clarify what we mean by 'motivation', given that this could be confused with aspects of psychology theory.

We also significantly rewrote the final paragraph of section 5 (starting line 424 in the prior version), with substantial input from the BOKU members of our team. Although we are not completely confident that we understand why the reviewer's mouth was open when reading, our guess is that the previous version was too simplistic, and did not cite appropriate references. We hope the new paragraph does a better job.

Many other small adjustments were made throughout, including minor vocabulary adjustments and text rearrangements. In general these are to improve clarity, accuracy and readability, and have no bearing on the results or conclusions. Most notably the 'catalytic' section was moved from part 3 to section 5.1. Additional changes include:

- Changed pedosphere to regosphere, which is more appropriate according to the description of terms in Huggett (2024)
- Added headings to the principles, and streamlined them (section 3.4)
- Combined the Tier 1 MEUTEC categories Deliberate neural restructuring and Experience oriented, to form 'Information, organization and neural interaction' (since they really aren't well distinguished) and expanded the discussion of how hard these are to categorize
- Separated the 'ambient context' category to be clearly unassociated with activities
- Accordingly we revised Figure 2, and simplified and rearranged it for legibility.
- Revised Figure 3 and tables to match adjusted categorization
- The supplement was updated, and reformatted to be tidy.

We hope that these largely-aesthetic improvements will be appreciated by the reader.

Warm regards,

Eric Galbraith, on behalf of coauthors